# Per-example Gradients: a New Frontier for Understanding and Improving Optimizers

Vincent Roulet [* 1]   Atish Agarwala [* 1]

## Abstract

When computing gradients, deep learning training algorithms typically treat the mini-batch as a fundamental unit — only returning batch-averaged gradients. Computing non-linear statistics of the mini-batch gradient distribution has traditionally been viewed as prohibitively expensive or requiring complex, custom implementations. We challenge this view by demonstrating that sequence-level architectures offer a natural testbed for prototyping algorithms based on per-example gradients. We show that staged programming languages like JAX enable generic manipulations of mini-batch gradient computations. We then build on Dangel et al. (2019) to derive implementations of specific per-example or per-token operations with negligible computational or memory overhead. Finally, we leverage our findings to re-examine two non-linear optimization operations. First, we analyze signSGD, showing that the optimal placement of the sign operation is critical to success and can be predicted via a simple signal-to-noise ratio argument. Second, we investigate per-example variations of the Adam preconditioner and find that, contrary to conventional wisdom, optimization is best served when the preconditioner is dominated by the mean squared of the gradient distribution rather than its variance. Overall our work shows that accessible per-example gradient information unlocks new avenues for algorithm analysis and design.

## 1. Introduction

The success of modern machine learning was enabled by efficient methods of computing gradients of loss functions through neural networks, primarily through the invention of reverse-mode automatic differentiation (AD) (Linnainmaa, 1976). With the advent of accelerators like GPUs, practitioners are now able to approximate the expected gradients of a model's loss function on a dataset by averaging gradients on large batches of data. This has allowed model and data complexity to grow rapidly while maintaining tractability of training algorithms (Hoffmann et al., 2022).

In recent years, there has been growing interest in accessing and using more complex information about the gradient distribution. For example, per-example gradient covariance statistics is useful for understanding deep learning; this information can be used to predict quantitative properties of loss trajectories at scale (Yin et al., 2018; McCandlish et al., 2018; Liu et al., 2020; Faghri et al., 2020; Gray et al., 2024; Qiu et al., 2025). There is practical interest in per-example gradient transformations as well. There is some evidence that optimizers which have access to per-example gradients can have better stability and predictability than typical optimizers (Wang & Aitchison, 2024). The growing interest in distributed optimizers like DiLoCo (Douillard et al., 2023) calls for study of optimizers that operate beyond just batch-averaged gradients (Zhang et al., 2023).

However, a key feature of reverse-mode AD is that it never stores the gradients of individual elements of a batch. This increases the memory efficiency of the process, but makes it impossible to answer questions about the distribution of per-example gradients, or allow for optimizers to depend on higher order moments of the gradient distribution. This leaves a vast part of training algorithm design space inaccessible for researchers, especially at large scales. There has been attempts to efficiently compute statistics per-example gradient norms (Kong & Munoz Medina, 2023; Bu et al., 2023), and to come up with ways to modify AD more generally to compute higher order moments of gradient distributions (Dangel et al., 2019; Schneider et al., 2021), but general development remains difficult.

Motivated by these examples, we study the technical challenges arising from computing generic gradient statistics. We show the following (Section 2):

- In some workloads, including transformers, simple

*Equal contribution [1]Google DeepMind. Correspondence to: Vincent Roulet <voulet@google.com>, Atish Agarwala <thetish@google.com>.

*Proceedings of the 43rd International Conference on Machine Learning*, Seoul, South Korea. PMLR 306, 2026. Copyright 2026 by the author(s).

automatic vectorization tools like `vmap` in JAX (Bradbury et al., 2018) allow for quick prototyping without incurring prohibitive overheads,

- Per-example or "per-token" gradient statistics may be implemented with a negligible memory and computational overhead for promising algorithms and metrics on specific architecture families.

Our results suggest that computing generic gradient statistics is not prohibitively expensive, and in some cases comes at virtually no overhead. We then use gradient statistics to improve our understanding of two optimization algorithms:

- **SIGNSGD** (Section 3.1). We study a per-element version of SIGNSGD and find that the optimal positioning of the sign operation is as late as possible in the processing chain — a phenomenon which can be understood using a simple signal-to-noise ratio analysis.
- **ADAM** (Section 3.2). We study ADAM variants which operate on per-example statistics like the second moment. We provide, to the best of our knowledge, the first true implementation of these algorithms at scale, confirming the predicted scaling properties with batch size. We show that preconditioners which depend on the mean squared train faster and more stably than those which depend on the variance, in contrast to conventional wisdom.

Altogether, our results suggest that studying per-example gradients/gradient transformations is tractable even in modern settings and opens a new dimension for understanding and improving training algorithms.

## 2. Accessing mini-batch gradient statistics

### 2.1. Objective and challenges

Deep learning pipelines generally consist of optimizing the parameters $\theta$ of a model $f$ as follows:

1. Fetch a mini-batch of samples $x_1, \ldots, x_B$ of size $B$:
   `fetch_batch(train_data)` $= x_1, \ldots, x_B$
2. Compute mini-batch loss in an automatic differentiation (AD) framework.
   `loss(params, batch)` $= \frac{1}{B} \sum_{i=1}^{B} f(\theta; x_i)$
3. Get the gradient of the mini-batch loss w.r.t. the parameters by gradient backpropagation.
   `grad(loss)(params, batch)` $= \nabla \frac{1}{B} \sum_{i=1}^{B} f(\theta; x_i)$
4. Feed the mini-batch gradient to an optimizer that returns directions along which parameters are updated.

As such, optimizers only have access to an estimate of the expectation of the gradients

$$\mathbb{E}[\nabla f(\theta; X)] \approx \frac{1}{B} \sum_{i=1}^{B} \nabla f(\theta; x_i), \qquad (1)$$

where $x_1, \ldots, x_B$ are assumed to be i.i.d. samples of $X$.

However, other statistics might be of interest for metrics, scientific exploration, and new algorithm design. Consider the problem of estimating the expectation of an arbitrary function $\phi$ of the gradient distribution using the mini-batch:

$$\mathbb{E}[\phi(\nabla f(\theta; X))] \approx \frac{1}{B} \sum_{i=1}^{B} \phi(\nabla f(\theta; x_i)). \qquad (2)$$

Statistics of interest include the element-wise variance $\mathbb{E}[(\nabla f(\theta; X) - \mathbb{E}[\nabla f(\theta; X)])^2]$, the sign $\mathbb{E}[\text{sign}(\nabla f(\theta; X))]$, or clipping $\mathbb{E}[\text{clip}(\nabla f(\theta; X))]$.

As a running example, we will consider MICROADAM—a variant of ADAM which uses the average element-wise square gradients over a batch rather than the squared average gradients for the preconditioner (Wang & Aitchison, 2024). Formally, denoting per-example gradients $g_i = \nabla f(\theta, x_i)$, the preconditioner becomes

$$\nu_{\text{micro}} := \frac{1}{B} \sum_{i=1}^{B} g_i^2, \text{ in lieu of } \nu_{\text{adam}} := \left( \frac{1}{B} \sum_{i=1}^{B} g_i \right)^2. \qquad (3)$$

The rest of the update rule remains the same. We will benchmark this algorithm against standard ADAM to evaluate our methods, and will analyze the algorithm in detail in Section 3.2.

With basic access to gradient oracle calls like `grad(loss)(params, sample)`, there are two simple ways to compute $\nu_{\text{micro}}$, both at the ends of a computation-memory trade-off.

1. **Memory intensive implementation.** Compute and square $\nabla f(\theta; x_1), \ldots, \nabla f(\theta; x_B)$ in parallel, then take the average. This requires $BP$ memory storage, with $P$ the dimension of the parameters, and can be prohibitive for some architectures and hardware.
2. **Computationally intensive implementation.** Accumulate the average $\nu_{\text{micro}}$ in a loop for $i = 1$ to $B$, computing the gradient, $\nabla f(\theta; x_i)$, square it, and update $\nu_{\text{micro}} \leftarrow [(i-1)\nu_{\text{micro}} + \nabla f(\theta; x_i)^2]/i$. This prevents using more memory storage than the one needed to compute two gradients. However, it requires $B$ more calls to a gradient oracle.

In the remainder of the section, we will show that there are opportunities to improve on these naive methods by detailed examination of the compute flow during AD.

### 2.2. Memory bottlenecks in deep learning

The memory required to compute statistics of the form (2) does not reside only in the final $B$ gradients $\nabla f(\theta; x_i)$. Automatic differentiation (AD) first checkpoints intermediate values in a forward pass. This leads to a simple fact that

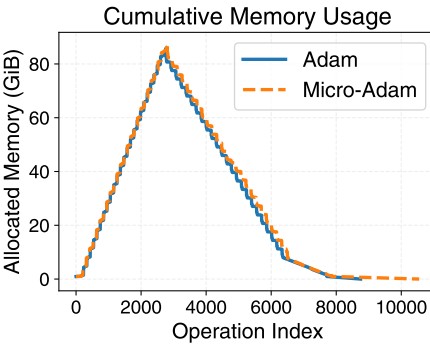

*Figure 1.* Memory footprint along program execution as reported by a code profiler of a train step. We compare the usual ADAM algorithm and its per-example variant, MICROADAM (Section 3.2) implemented in JAX using the automatic vectorization tool `vmap`. The peak memory corresponds to the accumulation of the memory during the forward pass of automatic differentiation necessary to compute gradients. We observe that the per-example variant may incur more operations (longer tail) that translate in longer execution time. But the peak memory is the same.

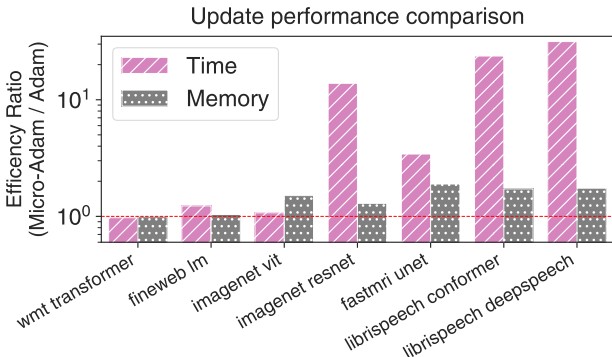

*Figure 2.* Relative efficiency of MICROADAM compared to ADAM in terms of time and memory costs across the workloads of the AlgoPerf Challenge (Dahl et al., 2023). On the three first workloads (transformers) time and memory costs overhead are modest and allow for quick prototyping of per-example methods. On the other workloads, memory requirements remain bounded to 2x the cost for Adam but may trade this limited increase with substantial time increase.

underpins potential opportunities to easily implement non-linear statistics like (2):

**Fact 2.1.** *For layers where the input size exceeds the parameter size, the memory typically reserved for input checkpoints can be repurposed to temporarily store $B$ individual gradients. This a priori enables the computation of nonlinear gradient statistics (2), such as $\nu_{\mathrm{micro}}$, without increasing peak memory usage.*

To better understand this, we review some standard layers using Einstein summation (einsum) notation:

1. **Dense layers in standard MLPs** ($\mathbf{D}, \mathbf{DF} \to \mathbf{F}$): Here, the parameter size, $DF$ is larger than the input size $D$. Storing individual gradients at a cost $BDF$ is larger than the cost of checkpointing inputs, $BD$, and is generally prohibitively expensive.
2. **Dense layers in transformers** ($\mathbf{LD}, \mathbf{DF} \to \mathbf{LF}$): Here as soon as the length $L$ is larger than the hidden dimension $F$, the cost of storing $B$ individual gradients is actually *less than the memory cost of the activations*!

Figure 1 shows the cost in memory incurred for ADAM or its per-example variant using $\nu_{\mathrm{micro}}$, MICROADAM, on a 1.2B transformer in Nanodo (Liu et al., 2024) using TPU v5e. It depicts a usual staircase shape arising from checkpointing (Blondel & Roulet, 2024, Figure 8.4). It confirms that, for a transformer, the peak memory remains almost unchanged. It also hints that the first layer could also benefit from all freed memory of later layers. Fact 2.1 only points out potential room for individual layers; the total cost depends on the global architecture and compilation optimizations that can trade-off memory and speed. A convolutional network may for example have some per-example-friendly layers like convolutions but also usual dense layers for which the cost

is not negligible.

Our analysis suggests that modern just-in-time compilers may be able to automatically identify and exploit this structure in some architectures. In order to test this hypothesis, we implemented MICROADAM across 7 workloads of the AlgoPerf challenge (Kasimbeg et al., 2025) using the JAX library's `vmap` vectorization capability, see Dahl et al. (2023, Section 4.3) for the workloads details. We compared the peak memory and time per step to ADAM (Figure 2). In the transformer model workloads for text or vision (first three), the increase in time or memory cost of the MICROADAM algorithm is modest, especially compared to the per-device batch size (see Appendix E.1). It illustrates that *experimentation with per-example methods on transformers is not prohibitively expensive*. On the other workloads we do not observe excessive increase in memory used (at most $2\times$) but this modest increase in memory may be paid by a non-negligible increase in computational cost as compilers may be able to automatically optimize for memory constraints with adequate fusions of some operations

### 2.3. Computational graph surgery

While `vmap` already provides us with a useful method across multiple workloads, we can improve even further by diving into the computational graphs generated by AD. Per-example information is preserved across most of the operations in the computation graph; the averaging of the gradients over the batch is generally the last operation (illustrated in Figure 3, detailed in Appendix B.1). If we parse the computational graph of the minibatch gradient, we can "inject" the desired computation $\phi$ to the individual gradients before averaging. For some operations $\phi$ like computing

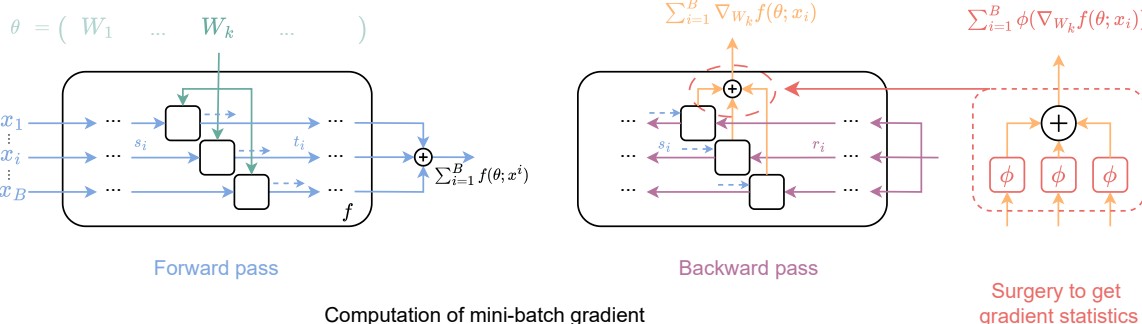

Forward pass                    Backward pass                    Surgery to get
                                                                 gradient statistics

Computation of mini-batch gradient

*Figure 3.* Computational graph of the mini-batch loss and the mini-batch gradient w.r.t. some intermediate weights. The forward pass has independent computational paths for each datapoint $x_i$ and intermediate activation $s_i$, and the weights are essentially broadcasted to each computational path before the final loss merges them (left). In the backwards pass the residuals $r_i$ move along the reversed computational paths and are similarly broadcast, and the merging of paths only happens at the end via sum reduction — the adjoint operation of the weight broadcasting. Computing gradient statistics of a function $\phi$ of the gradients can be done by injecting $\phi$ just before the final sum reduction.

$\nu_{\mathrm{micro}}$, and some layers reviewed below the injection can be done with negligible overhead.

**Dense layers in standard MLPs.** Take a dense layer in a standard MLP, parameterized by $W \in \mathbb{R}^{D \times F}$, which transforms a mini-batch of incoming activations $H \in \mathbb{R}^{B \times D}$ to outputs $Z \in \mathbb{R}^{B \times F}$. To compute the mini-batch gradient $\sum_b \nabla_W f(\theta; x_b) =: G \in \mathbb{R}^{D \times F}$ of the loss w.r.t. $W$, reverse mode AD performs a backward pass through transpose Jacobians across the network that computes the derivatives $\Lambda \in \mathbb{R}^{B \times F}$ of the loss w.r.t. the output of that layer. In einsum notations, the forward computation defining this layer and the computation leading to the gradient are

$$\begin{array}{cc} \mathrm{BF} \leftarrow \mathrm{BD}, \mathrm{DF} & \mathrm{DF} \leftarrow \mathrm{BD}, \mathrm{BF} \\ Z = H \quad W, & G = H \quad \Lambda, \end{array}$$

Both incoming activations $H$ and co-tangents $\Lambda$ are computed in independent computational paths illustrated in Figure 3. The mini-batch gradient $G$ agglomerates these paths in a sum of rank-one vectors $G = \sum_b h_b \lambda_b^\top$.

Computing the average elementwise squared gradients, can be done with a simple modification as we have $\sum_b \nabla_W f(\theta; x_b)^2 = \sum_b (h_b \lambda_b^\top)^2 = \sum_b h_b^2 \lambda_b^{2\top}$, i.e.,

$$\sum_b \nabla f(\theta; x_b)^2 =: \begin{array}{c} \mathrm{DF} \leftarrow \mathrm{BD}, \ \mathrm{BF} \\ S = H^{\odot 2} \ \Lambda^{\odot 2}, \end{array}$$

where $M^{\odot 2}$ denotes the elementwise square of a tensor $M$. Apart from squaring $H, \Lambda$, the result has the same memory cost, $O(BD + D^2)$, and computational complexity as computing the sum of gradients. It can take advantage of accelerators, and does not suffer from the $O(BD^2)$ memory cost of a naive memory-intensive implementation ( Section 2.1).

This computation of second moments for MLPs was first presented by Dangel et al. (2019, Appendix A.1). It can be

generalized to any operation $\phi$ that is factorable, i.e., such that $\phi(ab) = \phi(a)\phi(b)$ by using the rank-one nature of the gradients in this case. Examples of such functions include the sign function as well as any power $\phi(s) = s^\alpha$. In particular this means that any Taylor series can be efficiently implemented with this methodology. These algebraic manipulations are also similar to the "ghost-clipping" trick presented by Goodfellow (2015) that compute squared norms as the einsum contraction $(\|\nabla_W f(\theta; x_b)\|_2^2)_{b=1}^B N_b =: \leftarrow H_{b,d}, \Lambda_{b,f} N_b$.

**Dense layers in sequence-level architectures.** In sequence-level architectures like transformers, the forward computation and the computation leading to the gradient take the form

$$\begin{array}{cc} \mathrm{BLF} \leftarrow \mathrm{BLD}, \mathrm{DF} & \mathrm{DF} \leftarrow \mathrm{BLD}, \mathrm{BLF} \\ Z = H \quad W, & G = H \quad \Lambda, \end{array}$$

with $L$ denoting the length dimension of sequences fed to the model. Computing the average squared gradients amount to computing

$$\begin{array}{cc} \mathrm{BDF} \leftarrow \mathrm{BLD}, \mathrm{BLF} & \mathrm{DF} \leftarrow \mathrm{BDF} \\ M = H \quad \Lambda, & S = M^{\odot 2}, \end{array}$$

i.e., it requires first reducing over the length dimension, creating a $BDF$ tensor of per-example gradients, that is squared and then reduced. The leading complexity of the resulting operations remains $O(BLDF)$ but it has also an intermediate memory cost $O(BDF)$.

An alternative is to consider per-example "per-token" second moments, i.e., computing

$$\begin{array}{c} \mathrm{DF} \leftarrow \mathrm{BLD}, \mathrm{BLF} \\ S' = H^{\odot 2} \ \Lambda^{\odot 2} = \sum_{bl} h_{bl}^2 \lambda_{bl}^{2\top}, \end{array}$$

As for dense layers in MLPs, such an operation comes at a negligible overhead. In a transformer, tokens attend to

each-other in the attention layers, both in the forward and backward pass, because the transpose Jacobian of the attention block won't be separable along tokens. So the elements $\lambda_{bl}$ of the cotangents do not correspond to derivatives of the loss at token $l$ of sample $b$. Nevertheless the derived quantity can be used as proxy to capture relevant statistics. Note that different einsum contractions may also lead to per-sample, "per-token" norms as $N_{bl} \leftarrow H_{bld}^{\odot 2} \Lambda_{bld}^{\odot 2}$.

**Implementation** To implement these operations we may either define some custom derivative rules for the layers of interest as previously proposed by Dangel et al. (2019) in PyTorch (Paszke et al., 2019). Functional programming languages like JAX (Bradbury et al., 2018) or Functorch (Horace He, 2021) enable a more generic approach: a *computational graph surgery* that transforms the traced computation of a gradient. We provide code and benchmarking in Appendix B. For each operation of interest $\phi$, the method modifies the basic linear primitives in JAX. We note that a related approach was taken on a different problem (fast computation of NTK) in Novak et al. (2022). Such approaches modify basic framework operations, allowing automatic compatibility with downstream modules.

## 3. New perspectives from gradient statistics

In the remainder of the paper, we ask: what does access to this additional information actually buy us? We use the rapid prototyping `vmap` approach to study properties of optimization algorithms and their per-example counterparts, using methods and measurements enabled by our approach.

Many optimizers in deep learning can be written as processing batch gradients through a sequence of transformations $T_p, \ldots, T_1$ to define an update direction along which parameters are updated,

$$\theta_{t+1} = \theta_t - \eta T_p \circ \ldots \circ T_1(\text{avg\_grad}(\theta_t)). \quad (4)$$

Our methods unlock the potential for transformations that come after the gradient computation but before the averaging. Formally, we can change a training algorithm $T$ to a new algorithm $T'$ as follows:

$$T = T_2 \circ T_1 \circ \text{avg} \circ \text{grad} \rightarrow T' = T_2 \circ \text{avg} \circ T_1 \circ \text{grad},$$

where avg is the average over the gradients, and $T_1$ and $T_2$ are gradient transformations applied elementwise.

We focused on 2 settings: SIGNSGD and ADAM. We chose these settings for their relevance to practical optimization strategies, and since we know they can be implemented efficiently using the computational graph surgery approach. In order to provide insights relevant for modern machine learning, all experiments were performed on a 151M parameter decoder-only transformer language model, trained

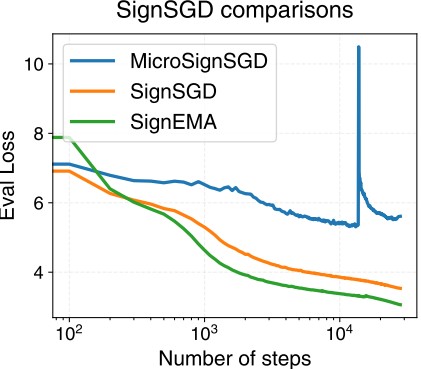

*Figure 4.* Learning curves for SIGNSGD variants at optimal learning rates, $\beta_1 = 0.9$. SIGNEMA has the best performance and MICROSIGNSGD has the worst performance. This suggests that the sign function needs to be applied as late as possible to prevent signal-to-noise ratio reduction for gradients of individual parameters.

on the C4 dataset (Raffel et al., 2020) using the Nanodo codebase (Liu et al., 2024). Unless otherwise specified, we use a batch size of 64, use a cosine learning rate schedule and a weight decay. Full experimental details can be found in Appendix E.

### 3.1. Where should you place the sign in SIGNSGD?

The optimizer SIGNSGD (Bernstein et al., 2018) has been well studied in various contexts, both from a theoretical perspective (Compagnoni et al., 2025; Balles et al., 2020; Xiao et al., 2025) but also as a practical optimizer (Zhao et al., 2025). The most basic SIGNSGD update rule is given by instantiating the typical optimizer (4) by applying the sign function elementwise to the average gradient. Theoretical work has shown that SIGNSGD is similar to RMSprop, effectively preconditioning by the square root of the diagonal of the Gauss Newton matrix (Xiao et al., 2025).

To make SIGNSGD a practical algorithm, practitioners add minibatching and momentum. A natural question is: what is the optimal order of operations? More concretely, we consider three operations: avg, which computes the empirical average of a set of minibatch gradients, EMA which takes an exponential moving average, and sign, applied to each parameter. The three algorithms we consider are:

$$\text{SIGNEMA} = \text{sign} \circ \text{EMA} \circ \text{avg}$$
$$\text{SIGNSGD} = \text{EMA} \circ \text{sign} \circ \text{avg}$$
$$\text{MICROSIGNSGD} = \text{EMA} \circ \text{avg} \circ \text{sign}$$

This covers all orderings of the 3 operations, since EMA must come after avg so that algorithms are independent of within-batch indexing.

MICROSIGNSGD is a new algorithm enabled by the per-

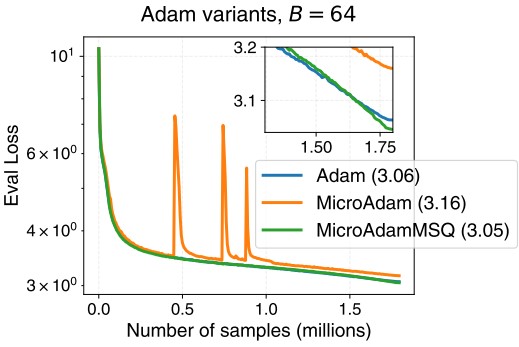

*Figure 5.* MICROADAM (orange) emphasizes variance information in preconditioner and generally trains less stably and more slowly than ADAM (blue), while MICROADAMMSQ (green, described in Section 3.2.3) emphasizes mean squared information and shows slight gains

example transforms. Here the sign operation is applied first, followed by the minibatch averaging, and then the momentum. SIGNEMA, a.k.a. signum, was found to be competitive with ADAM in training large transformer models (Zhao et al., 2025).

After optimizing learning rates, we find that SIGNEMA trains best, followed by SIGNSGD (Figure 4). Both train much better than MICROSIGNSGD. In addition to training more slowly, the MICROSIGNSGD curves are noisier (including a late-time training spike). Our results suggest that applying the sign function as late as possible results in the fastest, most stable training.

We hypothesize that MICROSIGNSGD fails in part because any stronger preconditioning is outweighed by the amplification of noise in the per-example transformations. The sign function reduces SNR for distributions with low SNR, and increases SNR for distributions with high SNR (Appendix C.1). This theoretical analysis is consistent with our observations that sign should be applied after the maximal amount of averaging — corresponding to the largest variance reduction possible for the object being transformed.

### 3.2. ADAM and per-example statistics

In this section, we consider variants of the preconditioner in ADAM on a per-example basis. We contrast the original ADAM algorithm that accumulates over time the squared average of gradients, $\nu_{\text{adam}}$ and its per-example variant MICROADAM, that uses $\nu_{\text{micro}}$ defined in (3). The quantities $\nu_{\text{adam}}$ and $\nu_{\text{micro}}$ can in-turn give us new insights on ADAM.

#### 3.2.1. ADAM, MICROADAM AND BATCH-SIZE SCALINGS

If the gradients $g_i = \nabla f(\theta; x_i)$ are i.i.d., we can compute the expected values of $\nu_{\text{adam}}$ and $\nu_{\text{micro}}$ as follows:

$$\mathbb{E}[\nu_{\text{adam}}] = \mu^2 + \sigma^2/B, \qquad \mathbb{E}[\nu_{\text{micro}}] = \mu^2 + \sigma^2, \quad (5)$$

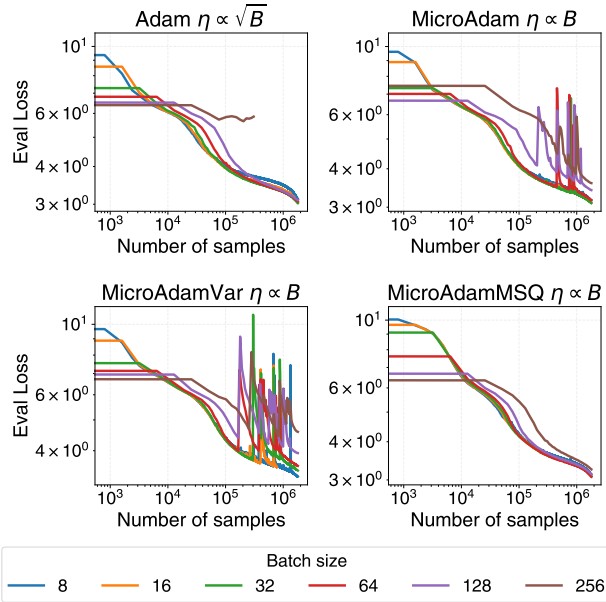

*Figure 6.* ADAM family variants trained at various batch sizes with their respective learning rate scaling rules. ADAM (top left) is trained with $\eta \propto \sqrt{B}$ and shows universal loss curves for intermediate batch size, but not for small or large batch size. MICROADAM (top right), MICROADAMMSQ (bottom left, Section 3.2.3), and MICROADAMVAR (bottom right, Section 3.2.3) all show universal scaling at small and intermediate batch sizes with $\eta \propto B$. ADAM family members with more $\sigma^2$ contribution to preconditioner suffer from stability issues, like MICROADAMVAR which *only* depends on $\sigma^2$. Learning rates are chosen to be close to optimal at $B = 64$.

where $\mu$ and $\sigma$ are the mean and standard deviation of the individual $g_i$.

For Adam, this observation is the basis of the *square root learning rate rule*. If $\mu = 0$, then the preconditioner scales as $B^{-1/2}$, which suggests that scaling the learning rate as $\eta \propto B^{1/2}$ causes the loss trajectories of models at different batch sizes to become similar when plotted versus the total number of samples (compared to SGD's rule $\eta \propto B$). This equivalence becomes exact in certain limits (Xiao et al., 2025). This heuristic also applies to the scaling of the optimal learning rate with batch size, for small batch sizes (Shallue et al., 2019). We provide a simple and intuitive derivation of the rule in Appendix D.1.

Wang & Aitchison (2024) point out that if $\mu \neq 0$, Equation 5 may have a crossover effect at large batch sizes where $\nu_{\text{adam}}$ goes from being variance dominated to mean squared dominated. They proposed MICROADAM— an optimizer where the preconditioner $\nu_{\text{adam}}$ of ADAM is replaced with the true second moment $\nu_{\text{micro}}$. They provided numerical evidence that this approach could lead to more universal learning curves for training ResNet18 on CIFAR10, using device parallelism to approximate $\nu_{\text{micro}}$ with a per-device batch-size of 25.

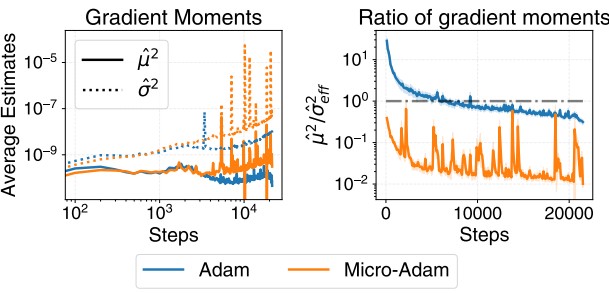

*Figure 7.* Estimator $\hat{\mu}^2$ is typically less than $\hat{\sigma}^2$ for both ADAM and MICROADAM, and quantities are similar for each algorithm at early times (left). However, the ratio of the $\mu^2$ and $\sigma_{\text{eff}}^2 = \sigma^2/B$ terms in $\nu_{\text{adam}}$ is greater than 1 at the start of training, and remains $O(1)$ until the end of training (right). In contrast, for MICROADAM the $\mu^2$ information is always significantly smaller than the $\sigma_{\text{eff}}^2 = \sigma^2$ term. Curves represent estimators using EMAs of $\nu_{\text{adam}}$ and $\nu_{\text{micro}}$, averaged over all parameters and all layers.

We took advantage of our methodology to evaluate MI-CROADAM without approximation in our language model setting. Our aim was to evaluate both the batch size scaling behavior, as well as the overall performance. We first tuned the learning rate at batch size $B = 64$ and found that MICROADAM trains poorly compared to ADAM with instabilities (Figure 5). The training spikes could be mitigated by gradient clipping, but the final eval loss values were worse than ADAM (3.06 vs 3.10, Appendix D.2).

We then evaluated the batch size scaling behavior of MI-CROADAM. Using the optimal learning rate $\eta^*$ at $B = 64$, we set the learning rate for other batch sizes using the predicted linear scaling rule $\eta = \eta^* B/64$. The resulting loss curves were plotted as a function of number of samples processed, where we see that for $B \in [8, 64]$ the learning curves follow the same universal form for much of training (Figure 6 top right). This is in contrast to the case of ADAM where we get universal curves for the square root learning rate rule $\eta = \eta^* \sqrt{B/64}$ (Figure 6 top left). We note that for ADAM, the smallest batch size $B = 8$ behaves in a non-universal way. This shows that the batch size scaling results previously approximated by Wang & Aitchison (2024) with "micro-batches" hold for a true implementation of MICROADAM as well. We note that the non-universality at larger $B$ is observed across all the algorithms and is due to the transition to the large learning rate regime where non-linear effects become prominent (Cohen et al., 2021; Agarwala & Pennington, 2024; Cohen et al., 2025).

The results shown in Figure 5 suggest that increasing the effect of the $\sigma^2$ term by switching from $\nu_{\text{adam}}$ to $\nu_{\text{micro}}$ is detrimental to neural network training on this workload. This suggests that information from $\mu^2$ may be more important to capture in the preconditioner. However, the square root learning rate scaling rule which holds for ADAM is motivated by the assumption that the variance dominates

the preconditioner and $\mu^2 \ll \frac{1}{B}\sigma^2$ (Appendix D.1). How do we resolve these seemingly inconsistent observations?

### 3.2.2. EXPECTATION AND VARIANCE OF GRADIENTS

We can measure $\mu^2$ and $\sigma^2$ directly by constructing unbiased estimators using $\nu_{\text{adam}}$ and $\nu_{\text{micro}}$:

$$\hat{\mu}^2 := \frac{1}{1 - B^{-1}}\left(\nu_{\text{adam}} - \frac{1}{B}\nu_{\text{micro}}\right), \text{ s.t. } \mathbb{E}[\hat{\mu}^2] = \mu^2,$$

$$\hat{\sigma}^2 := \frac{1}{1 - B^{-1}}\left(\nu_{\text{micro}} - \nu_{\text{adam}}\right), \qquad \text{s.t. } \mathbb{E}[\hat{\sigma}^2] = \sigma^2. \tag{6}$$

The latter is the traditional unbiased estimator of the variance using the sample variance. Therefore, if we can compute an estimate of $\mathbb{E}[\nu_{\text{adam}}]$ and $\mathbb{E}[\nu_{\text{micro}}]$ during training, we can directly compare $\mu^2$ and $\sigma^2$.

Using EMA estimators, we measured $\mu^2$ and $\sigma^2$ for both ADAM and MICROADAM. We used the same $\beta_2$ as the training algorithm so that $\nu_{\text{adam}}$ and $\nu_{\text{micro}}$ could be used for the training steps in their respective algorithms. We found that at early training times, ADAM and MICROADAM had similar values of $\hat{\mu}^2$ and $\hat{\sigma}^2$ (Figure 7, left). In both cases $\hat{\sigma}^2 > \hat{\mu}^2$, and the gap increased during training. In order to understand the relative importance of $\mu^2$ and $\sigma^2$, we define $\sigma_{\text{eff}}^2 = \frac{1}{B}\sigma^2$ for ADAM, and $\sigma_{\text{eff}}^2 = \sigma^2$ for MICROADAM. The ratio $\hat{\mu}^2/\hat{\sigma}_{\text{eff}}^2$ gives a better sense of the dependence of $\nu_{\text{adam}}$ and $\nu_{\text{micro}}$ on the two terms, and reveals that at early times, $\nu_{\text{adam}}$ has a larger contribution from $\mu^2$ than $\sigma^2$ (Figure 7, right). This is in contrast to MICROADAM which is dominated by $\sigma^2$ for most of training.

The trend of large $\mu^2$ in ADAM is consistent across models of different sizes trained with different $B$ (Figure 8). We found that for ADAM the ratio $\mu^2/\sigma_{\text{eff}}^2$ had a similar range across batch sizes when training with the square root scaling rule. We hypothesize that the dynamics induce a nontrivial relationship between $\mu^2$, $\sigma^2$ and $B$, which can preserve the usefulness of the square root scaling rule. We leave detailed characterization and a true causal explanation of this phenomenon to future work.

### 3.2.3. A NEW FAMILY OF ADAM METHODS

The results of the previous section suggest that $\mu^2$ information is better than $\sigma^2$ information in the ADAM preconditioner. This raises two additional questions: does a preconditioner based on $\sigma^2$ in (6) perform even worse than MICROADAM, and does one based on $\mu^2$ in (6) perform better than ADAM?

We implemented these optimizers, which we call MI-CROADAMVAR and MICROADAMMSQ respectively, using linear combinations of $\nu_{\text{adam}}$ and $\nu_{\text{micro}}$ as we had for the

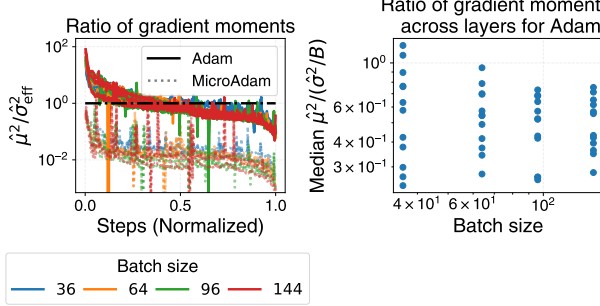

*Figure 8.* Ratios of $\hat{\mu}^2$ to $\hat{\sigma}_{\text{eff}}^2$ are consistent for models across different batch sizes with $\mu^2$ dominating at early times for ADAM (left). For ADAM, median ratio of $\hat{\mu}^2 / \frac{1}{B}\hat{\sigma}^2$ in different layers does not show much dependence on batch size (right), suggesting that the statistics of $\mu^2$ must have a nontrivial relationship with $\sigma^2/B$ to dominate the dynamics yet induce the square root learning rate heuristic.

measurements, i.e.,

$$\nu_{\text{MICROADAMVAR}} := \hat{\sigma}^2, \quad \nu_{\text{MICROADAMMSQ}} := \hat{\mu}^2,$$

for $\hat{\sigma}^2$, $\hat{\mu}^2$ defined in (6). Full pseudocodes of ADAM variants are presented in Appendix F. We found that indeed MICROADAMVAR had similar stability issues to and even worse performance than MICROADAM (Figure 6, bottom right). We also found that MICROADAMMSQ had a tendency to destabilize in only a few hundred steps, since the estimator $\mu^2$ in (6) is not guaranteed to be non-negative ($\nu_{\text{adam}}$ can be 0 for a finite sample, while $\nu_{\text{micro}}$ is always positive).

We solved the convergence issue by adding filtering to $\nu$, so that the preconditioner reads then $\sqrt{\text{ReLU}(\hat{\mu}^2)} + \epsilon$, with a larger $\epsilon = 10^{-6}$. Applying the ReLU only before applying the preconditioner maintains the unbiased nature of the estimator. This was combined with gradient clipping to obtain an algorithm which trains smoothly with slightly better performance than ADAM at batch size $B = 64$ (loss of 3.05 vs 3.06, Figure 5). It lagged slightly behind ADAM for most of training and then surpasses it at the end. As expected, MICROADAMMSQ shows universal curves over a large range of batch sizes with the scaling rule $\eta \propto B$ using the same methodology as for MICROADAM (Figure 6, bottom left, using $0.5\eta^* B$ where $\eta^*$ is optimal at $B = 64$). There was 20% more wall-clock time with our `vmap` implementation.

A natural question is: why is ADAM the best stable member of this extended ADAM family? We observe that the ADAM preconditioner is the unique linear combination of $\nu_{\text{adam}}$ and $\nu_{\text{micro}}$ that minimizes the $\sigma^2$ term while maintaining non-negativity. This suggests that improvements to ADAM along these lines would require emphasizing the $\mu^2$ information without instability due to small or negative contributions to the estimator.

Our experiments with MICROADAMMSQ demonstrated one mitigation strategy but others (such as averaging $\nu_{\text{micro}}$ and $\nu_{\text{adam}}$ across blocks of parameters) are worth exploring.

## 4. Discussion

One key finding of our methodological work is that the complexity of modern architectures and datasets actually *helps* methods that process gradient per-example, since the memory and compute bottlenecks are elsewhere in the AD calculations. This reflects a broader point that there are many potential opportunities to improve training algorithms by exploiting dimensions which are underutilized but not limiting in terms of memory or compute. Our computational graph surgery approach also demonstrates that further improving the implementation of these methods is possible in some cases. That approach reflects the flexibility that differentiable programming languages like JAX offer with their native tracing of programs. We can use this approach on any factorable functions, including sign and power functions, and indeed any linear combination of factorable functions. This gives us access to a very broad set of per-example statistics. Looking beyond gradients, we can imagine generating other AD-like computational graphs to efficiently compute per-example statistics. For example, it may be possible to compute second order statistics like Hessian-vector products or the diagonal of the Gauss-Newton using related approaches.

Our results show that transformers are particularly well suited to exploit per-example gradients. However, we acknowledge that for several other architectures per-example gradients may induce a non-negligible overhead. Weight-tying or pre-processing weights can still impede a simple vmap implementation or its computational graph surgery as explained in B.4.

Our experimental results show that access to per-example gradient statistics can dramatically improve our understanding of optimization in deep learning. Our most surprising result was that MICROADAM has stability and speed issues. By using the *measurements* enabled by our methods we were able to uncover evidence that the ADAM preconditioner is better when it is dominated by the mean squared rather than the variance — despite conventional wisdom like the explanation of the square root scaling rule. This is consistent with previous work showing that ADAM performs better with less noisy gradients (Kunstner et al., 2023), and overall suggests that constructing better estimators for this information might help improve ADAM. The success of square root scaling rule or variants like (Meterez et al., 2026) may have different roots that could further improve our understanding of schedules for example.

Altogether, these results suggest that per-example gradient

measurements and transformations are an exciting new area for optimization research. We can now test ideas about manipulating distributions of gradients in modern settings (Zielinski et al., 2020). We hope our methods will be used to develop new perspectives and paradigms in deep learning training algorithms in the near future.

## Impact Statement

This paper presents work whose goal is to advance the field of machine learning. There are many potential societal consequences of our work, none of which we feel must be specifically highlighted here.

## Acknowledgments

We deeply thank Keith Rush for sharing many discussions on the subject. Keith tremendously helped understand the innards of the computations in these models and was supportive of the idea from the start. We thank Mathieu Blondel for early conversations on the project and his support. We thank George Dahl for his insights on the memory costs. We thank Priya Kasimbeg for helping us evaluate the cost of per-example gradient transformations across the algoperf workloads. We thank Lechao Xiao for his careful review that helped improve the manuscript. Finally, we thank the reviewers for their thorough review and insightful feedbacks.

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

# A. Just-in-time compilation, tracing, and code optimization

A simple implementation of estimators like $\frac{1}{B}\sum_{i=1}^{B}\phi(\nabla f(\theta; x_i)) \approx \mathbb{E}[\phi(\nabla f(\theta; X))]$ would consist in (i) computing $B$ gradients, (ii) applying the non-linear operation $\phi$, (iii) averaging the result. In JAX, computing the average elementwise square gradient would look like this:

```python
import jax
import jax.numpy as jnp

B, d = 10, 4
fun = lambda w, x: jnp.sum(jnp.dot(x, w))
w, xs = jnp.ones((d, d)), jnp.ones((B, d))
indiv_grads = jax.vmap(jax.grad(fun), in_axes=(None, 0), out_axes=0)(w, xs)
avg_sq_grads = jnp.mean(indiv_grads**2, axis=0)
```

An immediate potential issue with this approach is the memory cost: computing $B$ gradients would require $B$ times the space necessary to store the parameters of the network. In many architectures such a cost is prohibitive.

However, not only such an implementation may be too naive, but it is also a naive view of how code is compiled in deep learning frameworks, like JAX (Bradbury et al., 2018) or Pytorch (Paszke et al., 2019). Such frameworks offer *just-in-time compilations (jit)* techniques that can tailor the compilations to a given backend and optimize the implementation to some extent. Below is the "jitting" for the average elementwise square.

```python
def compute_avg_sq_grads_(w, xs):
    indiv_grads = jax.vmap(jax.grad(fun), in_axes=(None, 0), out_axes=0)(w, xs)
    return jnp.mean(indiv_grads**2, axis=0)
compute_avg_sq_grads = jax.jit(compute_avg_sq_grads_)
```

Just-in-time compilation of a program $\mathcal{P}$ takes typical inputs of $P$, i.e., inputs with the shape and type that $\mathcal{P}$ will be run with, and lists all operations involved in $\mathcal{P}$ in order with the size and types of their inputs and outputs. Such a process is called *tracing* the program. In JAX, we get then access to the computational graph of $\mathcal{P}$ in the form of of a *jaxpr* that lists operations as JAX primitives in the topological order of the computational graph. Provided with such a computational graph in terms of some primitives, we can define a new computational graph to compute for example gradients of the outputs of $\mathcal{P}$ w.r.t. its inputs. Below is the jaxpr for the average elementwise square.

```python
print(jax.make_jaxpr(compute_avg_sq_grads_)(w, xs))
```

```
{ lambda ; a:f32[4,4] b:f32[10,4]. let
    c:f32[10,4] = dot_general[
      dimension_numbers=(([1], [0]), ([], []))
      preferred_element_type=float32
    ] b a
    _:f32[10] = reduce_sum[axes=(1,)] c
    d:f32[4] = broadcast_in_dim[broadcast_dimensions=() shape=(4,)] 1.0
    e:f32[4,10,4] = dot_general[
      dimension_numbers=(([], []), ([], []))
      preferred_element_type=float32
    ] d b
    f:f32[10,4,4] = transpose[permutation=(1, 2, 0)] e
    g:f32[10,4,4] = integer_pow[y=2] f
    h:f32[4,4] = reduce_sum[axes=(0,)] g
    i:f32[4,4] = div h 10.0
  in (i,) }
```

The computational graph leaves also room for optimizations and tailored implementations for given hardwares. For that, jaxprs are first converted in *high-level operations (HLO)* code, that are intermediate representations. They are independent of the framework (JAX, Pytorch, Tensorflow, etc...), so lower level than JAX. But they are also independent of the hardware, so higher level than targeted code for e.g. GPUs. Below is the hlo code for the average elementwise square.

```python
print(compute_avg_sq_grads.lower(w, xs).as_text())
```

```
module @jit_compute_avg_sq_grads_ attributes {mhlo.num_partitions = 1 : i32,
    mhlo.num_replicas = 1 : i32} {
```

```
nc.func public @main(%arg0: tensor<10x4xf32> {mhlo.layout_mode = "default"}) ->
    (tensor<4x4xf32> {jax.result_info = "", mhlo.layout_mode = "default"}) {
%cst = stablehlo.constant dense<1.000000e+00> : tensor<f32>
%0 = stablehlo.broadcast_in_dim %cst, dims = [] : (tensor<f32>) -> tensor<4xf32>
%1 = stablehlo.dot_general %0, %arg0, contracting_dims = [] x [], precision = [DEFAULT,
    DEFAULT] : (tensor<4xf32>, tensor<10x4xf32>) -> tensor<4x10x4xf32>
%2 = stablehlo.transpose %1, dims = [1, 2, 0] : (tensor<4x10x4xf32>) ->
    tensor<10x4x4xf32>
%3 = stablehlo.multiply %2, %2 : tensor<10x4x4xf32>
%cst_0 = stablehlo.constant dense<0.000000e+00> : tensor<f32>
%4 = stablehlo.reduce(%3 init: %cst_0) applies stablehlo.add across dimensions = [0] :
    (tensor<10x4x4xf32>, tensor<f32>) -> tensor<4x4xf32>
%cst_1 = stablehlo.constant dense<1.000000e+01> : tensor<f32>
%5 = stablehlo.broadcast_in_dim %cst_1, dims = [] : (tensor<f32>) -> tensor<4x4xf32>
%6 = stablehlo.divide %4, %5 : tensor<4x4xf32>
return %6 : tensor<4x4xf32>
```

HLO code can then be parsed to find simplifications or optimizations. Note that finding the high-level optimal implementation of a program may not be possible. The optimization of the implementation is done by parsing the graph with a series of simple rules (some available publicly on the OpenXLA project) that search for specific simplifications. The order in which the simplifications are done may even have an impact on the final implementation (Ganai et al., 2023).

In our case of interest, i.e., computing (2), a compiler may know the memory limits of the given hardware and accumulate the average by computing gradients one by one rather than computing all gradients at once. This would amount to *fuse* some operations in the graph (like the computations associated to e, f, g, h in the jaxpr above). This would implement a computationally intensive approach outlined in Section 2.1 but at the scale of the leaf of the computational graph. The trade-off memory vs time cost may further be optimized by the compiler by computing micro-batch of gradients at a time. Once the HLO code is optimized it is further converted in low-level code that further optimizes the code for the given hardware backend, see the documentation of the OpenXLA project `https://www.openxla.org/xla/` for further explanations.

To summarize, the actual implementation in frameworks like JAX involves many more moving parts than what a high-level python code looks like. An implementation using vmap typically already uses the trace of the computations of the original functions to modify appropriately the code for the needs of the user. For sequence-level models, the performance of the vmap approach only incurs a small computational overhead as illustrated in Figure 1. However, we have also evidence that a jitted vmap may not provide the best implementation in some other cases, see Figure 9. In general, a computational graph surgery can ensure efficient implementations. Some simplifications found in the computations of gradient statistics — like for computing the sum of square gradients in Section 2.3 — could also be implemented as parts of the optimization of the HLO code to further improve the native performance of vmap.

## B. Computational graph surgery

### B.1. From broadcasting weights, to reducing gradients, and injecting new statistics

We formalize Figure 3 to identify where, in the computational graph of the average gradients (1), is performed the mean reduction.

We analyze the loss $f(\theta; x_i)$ of a network with no shared weights schematized in Figure 3. We do not restrict ourselves to feed-forward networks. We examine the computation of the gradient of the sum of the losses, $\sum_{i=1}^{B} f(\theta; x_i)$. In the computation of the loss $f(\theta, x_i)$, we assume that the weight $W_k$ of the $k^{\text{th}}$ operation acts on the intermediate representation $s_i$ of the input $x_i$ through some bilinear operation $\ell$ to output

$$t_i = \ell(s_i, W_k).$$

For dense layers, $\ell$ is a vector-matrix product, for convolutional layers, it's a convolution, etc... Some sequence of operations later, the loss associated to this sample is computed and, as all samples in the mini-batch are processed, the average loss is computed, see the forward pass in Figure 3.

Computing $\nabla_{W_k} \sum_{i=1}^{B} f(\theta, x_i)$, i.e., the gradient of the sum of the losses w.r.t. the weight $W_k$, amounts then to computing

the gradient of a function of the form

$$W_k \mapsto \sum_{i=1}^{B} h_i(\ell_i(W_k)),$$

where $\ell_i = \ell(s_i, \cdot)$ and $h_i$ denotes subsequent operations on $s_i$, that may depend on the sample $x_i$ (if for example the sample actually consists of an input/output pair). This evaluation consists essentially in $B$ independent computational paths that are linked at the operational index $k$ by the weight $W_k$, and at the end by the reduction, see the forward pass in Figure 3.

Linear operations generally define a batch dimension along which inputs $s_i$ are treated by the same right operand $W_k$. This can be formalized by introducing the broadcast operation

$$\in : W \mapsto (W, \ldots, W),$$

that broadcasts the weights to the computation of the linear transformation of each input. We can then rewrite the computation of the gradient of the sum w.r.t. $W = W_k$ as

$$\nabla_W \left( \sum_{i=1}^{B} f(\theta; x_i) \right) = \nabla_W \left( \sum_{i=1}^{B} h_i(\ell_i(W)) \right) = \nabla \left( \oplus \circ h_{/\!/} \circ \ell_{/\!/} \circ \in \right)(W),$$

where

$$\oplus : (u_1, \ldots, u_B) \mapsto \sum_{i=1}^{B} u_i,$$

is the sum reduction, $\ell_{/\!/} : (W, \ldots, W) \mapsto (\ell_1(W), \ldots, \ell_B(W))$ denotes the execution of the linear operations on the replicas of the weights and the different inputs, and $h_{/\!/} : (t_1, \ldots, t_B) \mapsto (h_1(t_1), \ldots, h_B(t_B))$ denotes the rest of the computation.

Computing the gradient amounts to composing the Vector-Jacobian Products (VJP) of the operations, i.e., the adjoint[1] of their linear approximations on their inputs, in reverse order. Namely, following e.g. Blondel & Roulet (2024, Chapter 2),

$$\nabla \left( \oplus \circ h_{/\!/} \circ \ell_{/\!/} \circ \in \right)(W) = \partial \left( \oplus \circ h_{/\!/} \circ \ell_{/\!/} \circ \in \right)(W)^* 1 = \partial \in (W)^* \partial \ell_{/\!/}(V)^* \partial h_{/\!/}(t)^* \partial \oplus (u)^* 1,$$

where, for a function $f$, $\partial f(x)$ denotes its linearization around an input $x$, for a linear operator $a$, $a^*$ denotes its adjoint, and we used the shorthands $V = \in(W), t = \ell_{/\!/}(V), u = h_{/\!/}(t)$.

Since the broadcast and reduce-sum operations are already linear, their VJPs consist simply in their adjoint operations, i.e., $\in(W)^* = \in^*$ and $\oplus(V)^* = \oplus^*$. Moreover, one easily verifies that broadcast and reduce-sum are adjoint to each other, i.e., $\in^* = \oplus$, and $\oplus^* = \in$. One also easily observes that the linearizations $H_{/\!/} := \partial h_{/\!/}(t)^*, L_{/\!/} := \partial \ell_{/\!/}(V)^*$, of, respectively, $h_{/\!/}$ and $\ell_{/\!/}$ on their inputs, form independent paths of computations just as $h_{/\!/}$ and $\ell_{/\!/}$ did in their forward pass. The sum of gradients w.r.t. $W$ is then computed as

$$\nabla \left( \oplus \circ h_{/\!/} \circ \ell_{/\!/} \circ \in \right)(W_k) = \in^* \circ L_{/\!/}^* \circ H_{/\!/}^* \circ \oplus^* 1 = \oplus \circ L_{/\!/}^* \circ H_{/\!/}^* \circ \in 1.$$

We see —as illustrated in Figure 3— that the sum reduction happens theoretically as the last operation in the computation of the sum of the gradients. This is not due to the sum reduction of the losses (that is now a first broadcast operation in the computation of the gradient), but is due to the adjoint operator of the underlying broadcast of the weights in the forward pass.

To compute estimators of the form $1/B \sum_{i=1}^{B} \phi(\nabla f(\theta; x_i))$ as in (2), it suffices a priori to inject the nonlinear function just before the last operation as

$$\sum_{i=1}^{B} \phi(\nabla f(W; z_i)) = \oplus \circ \phi \circ L_{/\!/}^* \circ H_{/\!/}^* \circ \in 1.$$

In practice, we do not have direct access to a computation graph of the sum of the gradient of the form $\oplus \circ L_{/\!/}^* \circ H_{/\!/}^* \circ \in 1$. The reduction is performed as part of the VJP of $W_k \mapsto (\ell(s_1, W_k), \ldots, \ell(s_B, W_k))$. Diving into some specific forms show what opportunities exist for some linear operations as in dense layers, as presented in Appendix B.4. The generic viewpoint presented above lays down the foundations for a generic implementations of estimators (2) in a differentiable programming framework like JAX (Bradbury et al., 2018) by parsing the jaxpr of the gradient as explained in the next section.

---

[1]For a linear operator $a : \mathcal{E} \mapsto \mathcal{F}$ from a Euclidean space $\mathcal{E}$ to an Euclidean space $\mathcal{F}$, equipped with respective inner products $\langle \cdot, \cdot \rangle_{\mathcal{E}}$ and $\langle \cdot, \cdot \rangle_{\mathcal{F}}$, the adjoint of $a$ is the unique operator, denoted $a^*$ such that for any $u, v \in \mathcal{E} \times \mathcal{F}$, $\langle a(u), v \rangle_{\mathcal{F}} = \langle u, a^*(v) \rangle_{\mathcal{E}}$.

## B.2. Jaxpr surgery

As mentioned in Appendix A, JAX can and does trace the code, either to transform it (to provide the computational graph of the gradient) or to compile it efficiently. For our purposes, it means that we have access to jaxprs as the one below.

```python
import jax
import jax.numpy as jnp
import jax.random as jrd

B, d = 10, 4
fun = lambda w, x: jnp.sum(jnp.dot(x, w))
w, xs = jrd.normal(jrd.key(0), (d, d)), jrd.normal(jrd.key(1), (B, d))
loss = lambda w, x: jnp.sum(jnp.dot(x, w)**2, axis=-1)
sum_loss = lambda w, xs: jnp.sum(loss(w, xs))
sum_grad_jaxpr = jax.make_jaxpr(jax.grad(sum_loss))(w, xs)

print('Original jaxpr of sum of grads:\n', sum_grad_jaxpr)
```

```
Original jaxpr of sum of grads:
 { lambda ; a:f32[4,4] b:f32[10,4]. let
    c:f32[10,4] = dot_general[
      dimension_numbers=(([1], [0]), ([], []))
      preferred_element_type=float32
    ] b a
    d:f32[10,4] = integer_pow[y=2] c
    e:f32[10,4] = integer_pow[y=1] c
    f:f32[10,4] = mul 2.0 e
    g:f32[10] = reduce_sum[axes=(1,)] d
    _:f32[] = reduce_sum[axes=(0,)] g
    h:f32[10] = broadcast_in_dim[broadcast_dimensions=() shape=(10,)] 1.0
    i:f32[10,4] = broadcast_in_dim[broadcast_dimensions=(0,) shape=(10, 4)] h
    j:f32[10,4] = mul i f
    k:f32[4,4] = dot_general[
      dimension_numbers=(([0], [0]), ([], []))
      preferred_element_type=float32
    ] j b
    l:f32[4,4] = transpose[permutation=(1, 0)] k
  in (l,) }
```

Jaxprs encode a list of operations in the topological order of the computational graph. We can then encode computational graph surgeries by parsing the jaxpr to find where reduction happens as done below.

```python
import copy
import jax.extend as jex

def collect_reduce_ops(
    sum_grad_jaxpr: jex.core.Jaxpr,
) -> tuple[list[jex.core.JaxprEqn], list[jex.core.JaxprEqn]]:
  """Parse the jaxpr to collect last reduce operations before computing sum of
     gradients."""

  # Some primitives can safely be ignored
  invariant_primitives = ['transpose', 'reshape']

  # For this example, we simply treat 'dot_general' to showcase the overall
     implementation
  # Our library can handle other of the main reducing primitives like
  # 'reduce_sum', 'conv' etc...
  reduce_primitives = ['dot_general']

  reduce_ops = []

  # We track the reduce ops through the variables output when computing gradient
  grad_outvars = copy.copy(sum_grad_jaxpr.outvars)
```

```python
  for eqn in sum_grad_jaxpr.eqns[::-1]:
    primitive_name = eqn.primitive.name
    outvar = eqn.outvars[0]
    if outvar in grad_outvars:
      if primitive_name in invariant_primitives:
        # Add invar to list of variables tracked to find reduce ops
        invar = eqn.invars[0]
        grad_outvars.append(invar)
      if primitive_name in reduce_primitives:
        # Primitive found
        reduce_ops.append(eqn)
  return reduce_ops
```

Once we collected the operations we will change in the computational graph, we can simply parse the jaxpr as if it was normally evaluated and inject the transforms we want.

```python
from typing import Callable, Any

def jaxpr_surgery(
    sum_grad_jaxpr: jex.core.ClosedJaxpr,
    reinterpreter: Callable[[jex.core.JaxprEqn, jax.Array], jax.Array],
) -> Callable[..., jax.Array]:

  reduce_ops = collect_reduce_ops(sum_grad_jaxpr.jaxpr)
  def reinterpreted_sum_grad(
      params: list[jax.Array], *extra_fun_args: Any
  ) -> jax.Array:
    flat_params, params_treedef = jax.tree.flatten(params)
    flat_extra_fun_args = jax.tree.leaves(extra_fun_args)
    env = {}

    def read(var: jex.core.Var):
      if isinstance(var, jex.core.Literal):
        return var.val
      return env[var]

    def write(var: jex.core.Var, val: jax.Array):
      env[var] = val

    jaxpr = sum_grad_jaxpr.jaxpr
    consts = jaxpr.constvars

    args = flat_params + list(flat_extra_fun_args)

    jax.util.safe_map(write, jaxpr.invars, args)
    jax.util.safe_map(write, jaxpr.constvars, consts)

    for eqn in jaxpr.eqns:
      if eqn in reduce_ops:
        # Reinterpret the reduce operation
        invals = jax.util.safe_map(read, eqn.invars)
        ans = reinterpreter(eqn, invals)
      else:
        # Usual pass on the jaxpr graph
        subfuns, bind_params = eqn.primitive.get_bind_params(eqn.params)
        invals = jax.util.safe_map(read, eqn.invars)
        ans = eqn.primitive.bind(*subfuns, *invals, **bind_params)
      if eqn.primitive.multiple_results:
        jax.util.safe_map(write, eqn.outvars, ans)
      else:
        write(eqn.outvars[0], ans)

    flat_grads_like = jax.util.safe_map(read, jaxpr.outvars)
    return jax.tree.unflatten(params_treedef, flat_grads_like)
```

```
    return reinterpreted_sum_grad
```

The core and growing part of the library is then to implement reinterpreters that, for each possible reduction leading to a sum of gradients, implements the desired statistics. Below we present a simple and brief implementation for the sum of square gradients.

```
def sum_square_reinterpreter(eqn: jex.core.JaxprEqn, invals: tuple[jax.Array, ...]) ->
    jax.Array:
  if eqn.primitive.name == 'dot_general':
    lhs_contracting_dims, rhs_contracting_dims = eqn.params['dimension_numbers'][0]
    if len(lhs_contracting_dims) == len(rhs_contracting_dims) == 1:
      # This is the dense layer case operating on vectors
      # We can simply square the inputs and apply the original matrix product
      sq_invals = [inval**2 for inval in invals]
      sum_sq_sgrads = eqn.primitive.bind(*sq_invals, **eqn.params)
    else:
      # Reinterpreter can be extended for more generic cases of dot-general
      # as in sequence base model.
      raise NotImplementedError
    return sum_sq_sgrads
  else:
    # Reinterpreter can be extended for other operations like convolution.
    raise NotImplementedError()
```

We can then instantiate our desired "sum of square gradients" oracle and compare to the vmap implementation.

```
compute_sum_sq_grads = jaxpr_surgery(sum_grad_jaxpr, sum_square_reinterpreter)
print('Modified Jaxpr:\n', jax.make_jaxpr(compute_sum_sq_grads)(w, xs))

def compute_sum_sq_grads_via_vmap(w, xs):
  return jnp.sum(jax.vmap(jax.grad(loss), (None, 0))(w, xs)**2, axis=0)
print('Jaxpr of vmap implementation:\n',
    jax.make_jaxpr(compute_sum_sq_grads_via_vmap)(w, xs))

print(
  'Do implementations match?\n',
  jnp.all(jnp.equal(compute_sum_sq_grads(w, xs), compute_sum_sq_grads(w, xs)))
)

Modified Jaxpr:
 { lambda ; a:f32[4,4] b:f32[10,4]. let
   c:f32[10,4] = dot_general[
     dimension_numbers=(([1], [0]), ([], []))
     preferred_element_type=float32
   ] b a
   d:f32[10,4] = integer_pow[y=2] c
   e:f32[10,4] = integer_pow[y=1] c
   f:f32[10,4] = mul 2.0 e
   g:f32[10] = reduce_sum[axes=(1,)] d
   _:f32[] = reduce_sum[axes=(0,)] g
   h:f32[10] = broadcast_in_dim[broadcast_dimensions=() shape=(10,)] 1.0
   i:f32[10,4] = broadcast_in_dim[broadcast_dimensions=(0,) shape=(10, 4)] h
   j:f32[10,4] = mul i f
   k:f32[10,4] = integer_pow[y=2] j
   l:f32[10,4] = integer_pow[y=2] b
   m:f32[4,4] = dot_general[
     dimension_numbers=(([0], [0]), ([], []))
     preferred_element_type=float32
   ] k l
   n:f32[4,4] = transpose[permutation=(1, 0)] m
  in (n,) }
Jaxpr of vmap implementation:
 { lambda ; a:f32[4,4] b:f32[10,4]. let
   c:f32[10,4] = dot_general[
```

```
      dimension_numbers=(([1], [0]), ([], []))
      preferred_element_type=float32
    ] b a
    d:f32[10,4] = integer_pow[y=2] c
    e:f32[10,4] = integer_pow[y=1] c
    f:f32[10,4] = mul 2.0 e
    _:f32[10] = reduce_sum[axes=(1,)] d
    g:f32[4] = broadcast_in_dim[broadcast_dimensions=() shape=(4,)] 1.0
    h:f32[1,4] = broadcast_in_dim[broadcast_dimensions=(1,) shape=(1, 4)] g
    i:f32[10,4] = mul h f
    j:f32[10,4,4] = dot_general[
      dimension_numbers=(([], []), ([0], [0]))
      preferred_element_type=float32
    ] i b
    k:f32[10,4,4] = transpose[permutation=(0, 2, 1)] j
    l:f32[10,4,4] = integer_pow[y=2] k
    m:f32[4,4] = reduce_sum[axes=(0,)] l
  in (m,) }
Do implementations match?
 True
```

The code above can be extended to a library and benchmarked against vmap.

### B.3. Jaxpr surgery vs vmap

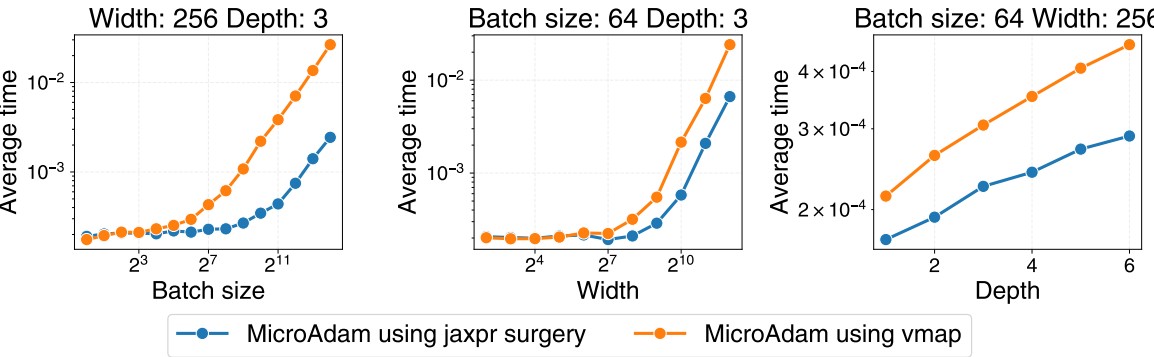

*Figure 9.* Execution times of MICROADAM with vmap or using a jaxpr surgery. The jaxpr surgery is generally faster and in particular scale much better with the batch size (note the log scale on the vertical time axis).

As explained in Appendix A, the native vmap transformation in JAX can benefit from optimizations at the HLO code level or even at a lower level of hardware. So even though the jaxprs of the vmap and the jaxpr surgery in the example above differ, the real performance of these methdos can only be judged in the final code execution.

To compare the jaxpr surgery and the vmap approaches, we implement the MICROADAM variant of ADAM that uses the average of square gradients rather than the square average of gradients (see Section 3.2). We implement it on MLPs with constant width (hidden dimension) across layers and measure the execution times of both approaches as depth, width or batch size vary, see Figure 9. For moderate batch sizes $B = 64$, the jaxpr surgery is moderately faster than the vmap implementation across widths and depths (Figure 9 middle and right panels) The jaxpr surgery can be an order of magnitude faster for batch sizes larger than $2^8 = 256$ (Figure 9 left panel). Both implementations follow similar trends with increasing batch size.

### B.4. Mini-batch gradient structure

**Explicit `einsum` notation.** Let $\mathcal{A}$ and $\mathcal{B}$ be tensors indexed by the sets of axis labels $\mathcal{I}_{\mathcal{A}}$ and $\mathcal{I}_{\mathcal{B}}$, respectively. The explicit einsum string defines a mapping $\mathcal{I}_{\mathcal{A}}, \mathcal{I}_{\mathcal{B}} \to \mathcal{I}_{\mathcal{C}}$ to produce an output tensor $\mathcal{C}$ indexed by $\mathcal{I}_{\mathcal{C}}$. Mathematically, the

elements of the output tensor are computed as:

$$\mathcal{C}_c = \sum_{k \in (\mathcal{I}_\mathcal{A} \cup \mathcal{I}_\mathcal{B}) \setminus \mathcal{I}_\mathcal{C}} \mathcal{A}_a \mathcal{B}_b$$

where $c$, $a$, and $b$ denote specific index assignments for the sets $\mathcal{I}_\mathcal{C}$, $\mathcal{I}_\mathcal{A}$, and $\mathcal{I}_\mathcal{B}$. In this formulation, the tensor product is summed strictly over the set of contraction indices $k$, which are defined as exactly those indices appearing in the input sets but explicitly omitted from the output set $\mathcal{I}_\mathcal{C}$.

As a concrete example, consider the forward pass of a dense layer in a standard MLP, mapped by the explicit signature $bd, df \to bf$. Here, the input index sets for activations $H$ and weights $W$ are $\mathcal{I}_H = \{b, d\}$ and $\mathcal{I}_W = \{d, f\}$, and the output index set for $Z$ is $\mathcal{I}_Z = \{b, f\}$. The contraction index set is therefore evaluated as $\{b, d, f\} \setminus \{b, f\} = \{d\}$, which rigorously recovers the standard batched matrix multiplication over the hidden dimension $d$:

$$Z_{b,f} = \sum_d H_{b,d} W_{d,f}.$$

**Tensor contractions.** We keep the framework outlined in B.1, i.e., we consider networks where weights define a linear operation on intermediate inputs and output $t_i = \ell(s_i, W_k)$. Here, we generalize the matrix-vector product case presented in Section 2.3 and consider generic tensor contractions.

*Forward operation.* Dropping the index $k$ of the operation in the weight $W = W_k$, and denoting $T = (t_1, \ldots, t_B), S = (s_1, \ldots, s_B)$, the batch operation reads $T = \ell(S, W)$ where $T, S, W$ are all multi-axes arrays. The tensor contraction consists in summing products of $S$ and $W$ along a set $K$ of *contraction* axes, and doing this for all "free indexes" of $S$ and $W$. Formally, following Einstein summation notations we get, for any $i_1, \ldots, i_m$ and $j_1, \ldots, j_n$ taken along a set of axes $I$ and $J$ respectively,

$$T^{i_1, \ldots, i_m}_{j_1, \ldots, j_n} = \sum_{k_1, \ldots, k_p} S^{i_1, \ldots, i_m}_{k_1, \ldots, k_p} W^{k_1, \ldots, k_p}_{j_1, \ldots, j_n}, \tag{7}$$

with $m = |I|, n = |J|, p = |P|$. The axes $I$ along which $i_1, \ldots, i_m$ are taken can be seen as the "data parallel" indexes along which the content of the intermediate representations $S$ is processed. In particular, $i_1$ will denote the mini-batch axis along which we compute the independent computational paths of the losses. The axes $J$ along which $j_1, \ldots, j_n$ are taken can be seen as the "new dimensions" of the intermediate representations.

*Examples.* For a dense layer operating on vectors, we have $m = n = p = 1$, and for all $i, j$,

$$T^i_j = \sum_k S^i_k W^k_j.$$

There, $i$ is the index of the batch, $k$ sums over the hidden dimension common to the input and the matrix, and $j$ indexes the dimension of the new representations.

For dense layers operating on sequences, we have $m = 2, n = p = 1$, and for all $i, l, j$,

$$T^{il}_j = \sum_k S^{il}_k W^k_j.$$

There $i, j, k$ play the same role as above, but the operation also has a "length" axis with $l$ denoting the index in the sequence on which the operation is applied.

For convolutions, several implementations pass by folding/unfolding the images as explained by Bu et al. (2022). Essentially, for an input image $s_i \in \mathbb{R}^{H \times W \times C}$ convolved with a filter $w \in \mathbb{R}^{K \times K \times C \times C}$, the image is unfolded into $\sigma_i \in \mathbb{R}^{HW \times CK^2}$, and the weights into $\omega \in \mathbb{R}^{CK^2 \times C}$ such that the matrix product $\tau_i = \sigma_i \omega \in \mathbb{R}^{HW \times C}$ corresponds to the convolution at each output index (variable heights, width, channels, and kernel sizes can be treated with the same logic, see Bu et al. (2022)). The output image is obtained by folding $\tau_i$ into $t_i \in \mathbb{R}^{H \times W \times C}$. For a batch of examples, the core operation remains a tensor contraction

$$\tau^{il}_j = \sum_k \sigma^{il}_k \omega^k_j,$$

where $i$ is still the index in the batch, while $l$ is the spatial position after unfolding, $k$ denotes the index of the receptive fields of the image and the filter concatenated along the channels of the inputs, and $j$ denotes the output channel dimension.

*Mini-batch gradient computation.* During backpropagation, as explained in Appendix B.1, the gradients of the loss w.r.t. $T$ are computed giving a multi-index array $R$ sharing the same indexes as $T$. One verifies then that the adjoint $\ell(S, \cdot)^*$ of the tensor contraction, give a final sum of gradients as $\sum_{i=1}^{B} \nabla_W \ell(\theta; x_i) = G$ where

$$G_{j_1,\ldots,j_n}^{k_1,\ldots,k_p} = \sum_{i_1,\ldots,i_m} S_{k_1,\ldots,k_p}^{i_1,\ldots,i_m} R_{j_1,\ldots,j_n}^{i_1,\ldots,i_m}, \tag{8}$$

is computed as a tensor contraction over the free axes $I$ of $S$ and $R$.

*Opportunities in vectorized intermediate representations.* For dense layers operating on vectors the final operation (8) reads

$$G_j^k = \sum_i S_k^i R_j^i,$$

that can be written as a sum of rank-one vectors. As explained in 2.3 we can exploit this structure for factorable operations like the elementwise square.

*Potential challenges in sequence-level operations.* For dense layers operating on sequence of vectors, the final operation (8) reads

$$G_j^k = \sum_i \sum_l S_k^{i,l} R_j^{i,l}.$$

In other words, in this case, the per-sample gradients are not rank-one vectors and require a sum across the length of the sequence. We cannot then simply square $S$ and $R$ to obtain for example the sum of square gradients.

One may think that $\sum_i \sum_l (S_k^{i,l})^2 (R_j^{i,l})^2$ represent sum of square gradients per-sample per-token; however for architectures like transformers this is not a well-posed quantity since the computations of per-token losses do not define independent computational paths (the activation heads mix the contribution of each token). That sum instead represents a per-sample per-activation gradient.

**Affine operations.** Note that affine operations (like adding an offset) can essentially be cast in terms of tensor contractions between the offset $b$ and some constant vector of ones. We can then make similar observations as in the generic tensor case: if the offset is applied on several "data parallel" axes (like both the batch and some length axis) then the gradient is obtained through several reductions. If there is only one axis, we can compute e.g. average square of gradients for offsets easily too.

**Case of preprocessed weights.** For some operations weights are preprocessed before being applied to the inputs. For example, weight matrices may be orthonormalized before being applied (Mhammedi et al., 2017). Operations on the inputs take then the form

$$\ell(s_i, W_k) \quad \text{for } W_k = p(V_k),$$

where $V_k$ are actually the weights we are optimizing for. Such situations generally cover cases where the weights are not applied linearly on inputs, as it can be the case for the scaling of a normalization layer.

If the weights are preprocessed, following the perspective taken in Appendix B, the mini-batch gradient with respect to $V_k$ read

$$\nabla \left( \oplus \circ h_{/\!/} \circ \ell_{/\!/} \circ \in \circ p \right)(V_k) = P^* \circ \oplus \circ L_{/\!/}^* \circ H_{/\!/}^* \circ \in 1,$$

where $P = \partial p(V_k)$ is the linearization of $p$ on $V_k$.

To average a function $\phi$ of the gradients, we need to first apply the adjoint of the preprocessing on each individual gradients and then average. In other words, the resulting computations would be

$$\sum_{i=1}^{B} \phi(\nabla_{V_k} f(\theta; x_i)) = \oplus \circ \phi \circ P_{/\!/}^* \circ L_{/\!/}^* \circ H_{/\!/}^* \circ \in 1,$$

where $P_{/\!/}^*$ applies the adjoint $P^*$ on all incoming individual gradients computed through $L_{/\!/}^* \circ H_{/\!/}^* \circ \in 1$. Injecting the non-linear operation $\phi$ requires then to modify the adjoint $P^*$ to operate on individual gradients, and we may not use some simplifications like applying $\phi$ to the inputs $s_i$, $r_i$ of the reduction as explained in Section 2.3.

**Tied weights.**  Tied weights, i.e., weights that are applied at different operations, may still be treated by a jaxpr surgery, albeit at a potentially non-negligible cost.

Consider for example an RNN whose weight $W$ is shared among the time-steps $k$. The resulting mini-batch gradient is then summed both along the batch axis and along the operations in the computational graph, and take the form $\sum_i \nabla_W f(\theta, x_i) = \sum_i \sum_k s_k^i (r_k^i)^\top$ for $s_k^i$ the intermediate representations of $x_i$ along the time steps $k$ and $r_k^i$ the gradients with respect to the output of each application of $W$.

When computing the mini-batch gradient, the sum over the batch is computed at each index and the sum over the time steps is accumulated during the backward pass. Namely, the gradient of the mini-batch is naturally computed as $\sum_k \sum_i s_k^i (r_k^i)^\top$.

Getting access to the individual gradients require then to remove the sum over the batch in the computational graph of the gradient to back-propagate the $B$ individual gradients up to the first time step. At that point, the $\phi$ operation can be applied and a final sum over $i$ can be performed.

Note that, if the RNN processes vectors, the intermediate gradients $s_k^i (r_k^i)^\top$ are rank-one vectors whose nature can be exploited. In particular, for any operations that factor over the rank-one structure such as taking the square, we can bookkeep the decomposition in rank-one vectors to potentially avoid some memory overload.

## C. SIGNSGD

### C.1. The sign **function and signal to noise ratio**

Applying the sign function on a random variable $X$ can dramatically change the signal to noise ratio (SNR) as defined by $\mathrm{SNR}_X \equiv |\mu_X|/\sigma_X$. The SNR of $\mathrm{sign}(X)$ is

$$\mathrm{SNR}_{\mathrm{sign}(X)} = \frac{|\mathbb{E}[\mathrm{sign}(X)]|}{\sqrt{\mathrm{Var}[\mathrm{sign}(X)]}} = \frac{|2p-1|}{2\sqrt{p(1-p)}}, \tag{9}$$

where $p$ is the probability that $X > 0$.

Note that the ratio $\mathrm{SNR}_X / \mathrm{SNR}_{\mathrm{sign}(X)}$ depends heavily on the distribution — $\mathrm{SNR}_{\mathrm{sign}(X)}$ is sensitive to the details of tails, and imbalance like discrepancy between the median and mean. Regardless, it is informative to analyze some simple cases.

Consider the case where $X \sim \mathcal{N}(r, 1)$. We have $\mathrm{SNR}_X = r$. For $0 < r \ll 1$, we have:

$$p = \frac{1}{2} + \frac{r}{\sqrt{2\pi}} + O(r^2), \quad \mathrm{SNR}_{\mathrm{sign}(X)} = \frac{r}{\sqrt{2\pi}} + O(r^2). \tag{10}$$

Therefore the sign operation reduces the SNR in this limit. In general, for distributions whose PDF can be defined as differentiable functions $\rho(x - r)$ for some small parameter $r$, where $\rho(x)$ has median at 0, we have:

$$\mathrm{SNR}_{\mathrm{sign}(X)} = \rho(0)r + O(r^2). \tag{11}$$

If the distribution $\rho(x)$ has 0 mean as well, then the distribution $\rho(x - r)$ has mean $r$. The change in SNR due to the sign function can be written as:

$$\frac{\mathrm{SNR}_{\mathrm{sign}(X)}}{\mathrm{SNR}_X} = \frac{\rho(0)}{\sigma_X} + O(r^2) \tag{12}$$

This suggests that for narrower distributions (in the sense of $\rho(0)/\sigma_X$), sign has a worse effect on SNR than for broader distributions — which is consistent with general intuitions and observations about the sign function in optimization.

In the opposite limit where $r \gg 1$ we have:

$$p = 1 - \frac{\exp(-r^2/2)}{r}(1 + O(r^{-2})), \quad \mathrm{SNR}_{\mathrm{sign}(X)} = \sqrt{r}\exp(r^2/4). \tag{13}$$

Here SNR is improved by sign if $r > \sqrt{2\log(r)}$.

Our analysis suggests that application of sign can reduce SNR when SNR is already low, and increase it when SNR is high, which means one must be careful about where to apply it in gradient processing. If gradients are near-Gaussian, it suggests that it is better to apply sign after as much averaging as possible to reduce the SNR of the random object that is passing through sign. This is consistent with the observation that SIGNEMA was the best performing algorithm, and MICROSIGNSGD was the worst.

# D. ADAM variant details

## D.1. Argument for the square root scaling rule

In this section we present a basic argument for the proposed $\eta \propto \sqrt{B}$ scaling heuristic that is often used for ADAM. We start with the scaling rule $\eta \propto B$ for SGD. The basic setup is as follows: we consider a series of gradients $g_t$ presented in a training loop. (We will focus on the case of a single parameter at the time but the argument generalizes immediately to multiple parameters.) We assume that the gradients are sampled i.i.d. from a distribution with mean $\mu$ and variance $\sigma^2$.

Consider taking $T$ steps of SGD with batch size $B$ with learning rate $\eta$. The mean and the variance of the total update $u_T(\eta, B)$ is given by:

$$\mathbb{E}[u_T(\eta, B)] = T\eta\mu, \ \text{Var}[u_T(\eta, B)] = T\eta^2 \frac{\sigma^2}{B}. \tag{14}$$

We can then ask the following question: is there some learning rate scaling rule such that the first two moments are identical for a fixed number of samples $K \equiv TB$? That is, we want:

$$\mathbb{E}[u_{K/B}(\eta(B), B)] = m(K), \ \text{Var}[u_{K/B}(\eta(B), B)] = V(K). \tag{15}$$

The answer is linear scaling of learning rate with batch size. Selecting $\eta = aB$, we have:

$$\mathbb{E}[u_{K/B}(aB, B)] = aK\mu, \ \text{Var}[u_{K/B}(aB, B)] = a^2 K \sigma^2. \tag{16}$$

If we think of modeling SGD as a random stochastic process, this ensures that we have two processes with equal statistics in a sample-to-sample comparison.

We can now provide a similar analysis for ADAM. We ignore momentum for now and model the ADAM update distribution as follows: we assume that the gradient distribution is stationary, and the preconditioner well-approximates the second moment of this distribution. That is, the preconditioner magnitude is $\mu^2 + \sigma^2/B$. Note that the batch size appears in this calculation.

Under these assumptions, the modified gradients $\tilde{g}_t$ have central moments

$$\mathbb{E}[\tilde{g}_t] = \frac{\mu}{\sqrt{\mu^2 + \sigma^2/B}}, \ \text{Var}[\tilde{g}_t] = \frac{\sigma^2}{\mu^2 + \sigma^2/B}. \tag{17}$$

The denominators come from the assumption that the preconditioner is well-captured by the second moment $\mathbb{E}[g_t^2]$.

The update central moments are

$$\mathbb{E}[u_T(\eta, B)] = T\eta \frac{\mu}{\sqrt{\mu^2 + \sigma^2/B}}, \ \text{Var}[u_T(\eta, B)] = T\eta^2 \frac{\sigma^2}{B\mu^2 + \sigma^2}. \tag{18}$$

We can ask the same question about finding a learning rate scaling rule that causes moments to match for equal number of samples. In general this would involve a complicated rule depending on the gradient statistics; however this simplifies if $\mu^2 \ll \sigma^2/B$. In this case we have:

$$\mathbb{E}[u_T(\eta, B)] \approx T\eta\sqrt{B}\frac{\mu}{\sigma}, \ \text{Var}[u_T(\eta, B)] \approx T\eta^2. \tag{19}$$

Here the *square root scaling rule* $\eta \propto a\sqrt{B}$ gets us

$$\mathbb{E}[u_{K/B}(aB, B)] = aK\frac{\mu}{\sigma}, \ \text{Var}[u_{K/B}(aB, B)] = a^2 K, \tag{20}$$

independent of $B$. Interestingly this is a normalized version of the moments for SGD. The other limit is $\mu^2 \gg \sigma^2/B$. Then we have

$$\mathbb{E}[u_T(\eta, B)] \approx T\eta, \ \text{Var}[u_T(\eta, B)] \approx T\eta^2 \frac{\sigma^2}{B\mu^2}. \tag{21}$$

This leads to a linear scaling rule $\eta \propto aB$:

$$\mathbb{E}[u_{K/B}(aB, B)] = aK, \ \mathrm{Var}[u_{K/B}(aB, B)] = a^2 K \frac{\sigma^2}{\mu^2}, \tag{22}$$

unlike the small $\mu^2$ limit.

The square root scaling rule can be made more formal in various stochastic differential equation (SDE) limits of training dynamics. Currently such limits have been formally derived for SIGNSGD and not ADAM, but the same square root learning rule applies in both cases. In such a limit one can derive an SDE in continuous time whose dynamics can be mapped onto the discrete stochastic dynamics such that averages of observables match between the two descriptions, with error shrinking with some small parameter (learning rate in the approach of Compagnoni et al. (2025), inverse dimension in Xiao et al. (2025)).

In our training setup, we can see that the square root learning rule induces similar learning curves in terms of number of samples over some range of $B$ (Figure 10). We first found the best learning rate $\eta^*_{B_0}$ for $B_0 = 64$. We then use the square root scaling rule to generate learning rates for other batch sizes with $0.5\eta^*_{B_0}$ (Figure 10, left), and $\eta^*_{B_0}$ (Figure 10, right). We see that there is a crossover from small $B = 8$ to the larger batch sizes where the learning curve is quite different. From $B = 16$ to $B = 128$ the intermediate to late time learning curves are similar, while for larger batch sizes (and therefore learning rates) the curves become non-universal and eventually diverge. Smaller learning rates show better agreement as predicted by the theory; the heuristic (and the SDE equivalents) break down if the gradient distribution changes over a small number of steps. This can happen because of effects like feature learning, or even more simply by making significant progress towards the objective.

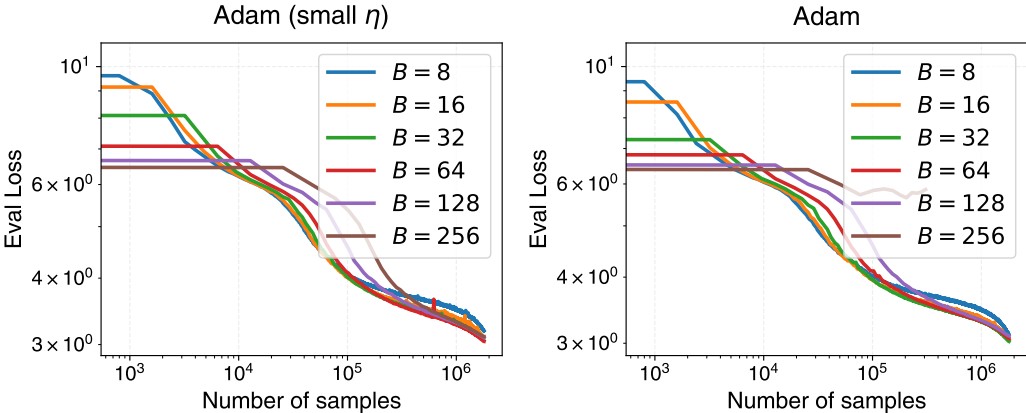

*Figure 10.* Square root learning rate scaling provides similar learning curves for a wide range of $B$ in transformer language model training. Small batch size ($B = 8$) shows crossover behavior. Large batch size leaves the universal regime. Learning rate at $B = 64$ was set to half the optimal rate in the left plot, and at the optimal rate in the right plot. Smaller learning rates show better universal behavior.

The observation that the square root rule works in practice suggests that $\mu^2 \gg \sigma^2/B$. However, our experiments suggest that $\mu^2 \sim \sigma^2/B$ across various batch sizes, preserving the square root scaling rule without a variance-dominated preconditioner. Understanding this mystery fully is left as an open question.

### D.2. Stabilizing variance-dominated ADAM variants with gradient clipping

We found that gradient clipping could stabilize both MICROADAM and MICROADAMVAR with appropriate choice of clipping thresholds. For $B = 64$ we swept clipping thresholds by factors of $10$ and found that a threshold of $10^{-2}$ removed most training spikes and led to the best eval metrics at the end of training. We then repeated the batch size sweep experiments from Figure 5 to confirm that the learning curves across batch sizes were similar and that training spikes were removed at most scales (Figure 11). However the clipping made the correspondence between the different batch sizes worse, and the final eval losses were still worse than ADAM (3.10 for MICROADAM, 3.20 for MICROADAMVAR, versus 3.06 for ADAM).

Indeed, the fact that MICROADAMVAR was the worst algorithm even after fixing stability issues further suggests that emphasizing the variance information in the ADAM preconditioner is generally detrimental for training.

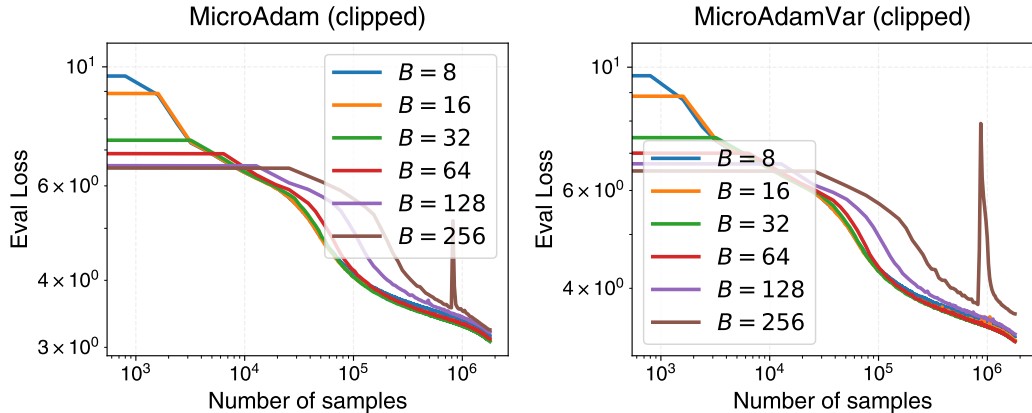

*Figure 11.* Training spikes in MICROADAM (left, best eval 3.10) and MICROADAMVAR (right, best eval 3.20) can be mitigated with strong gradient clipping with threshold $10^{-2}$. However these methods still have worse final eval loss compared to ADAM (3.06). This provides further evidence that emphasizing the variance term of the ADAM preconditioner leads to worse training outcomes.

## D.3. ADAM $\mu^2$ and $\sigma^2$ statistics

In order to further understand the consistency of the square root learning rule with our observation that $\mu^2$ seems to dominate the preconditioner at early times, we measured $\mu^2$ and $\sigma^2$ using the estimators from 6 for models trained with varying batch size. Plotting the ratio of $\hat{\mu}^2$ to $\hat{\sigma}^2_{\text{eff}}$ ($\sigma^2/B$ for regular ADAM, $\sigma^2$ for MICROADAM) we see that at all batch sizes we studied, $\mu^2$ dominates at early times for ADAM (Figure 8, left). The ratios and their dynamics were somewhat consistent even for ADAM, where $\hat{\sigma}^2_{\text{eff}}$ has explicit dependence on $B$.

To further quantify this, we took the median ratio for ADAM for each layer at each batch size, and found that the range of values was consistent across batch sizes (Figure 8, right). This is consistent with the idea that indeed the statistics of $\mu^2$ must have a nontrivial relationship with $\sigma^2/B$ to dominate the dynamics yet induce the square root learning rate heuristic. More study across a broader set of batch sizes and architectures is needed to firmly establish this phenomenon.

## E. Experimental details

### E.1. Testing per-example methods across workloads

For the experiments on the Algoperf codebase, we implemented MICROADAM in JAX using vmap via the Algoperf API. We trained on $2 \times 4$ slices of TPU v5e. We removed batchnorm from any workloads which had it since this normalization is incompatible with a `vmap` based approach. All workloads were run with the competition configurations. For convenience, the batch sizes (raw and per device) for each workload group can be found in Table 1.

| Workload name | Batch size | Batch size per device |
|---|---|---|
| fastmri unet | 32 | 4 |
| fineweb lm | 64 | 8 |
| imagenet (all) | 1024 | 128 |
| wmt transformer | 128 | 16 |
| librispeech (all) | 256 | 32 |

*Table 1.* Algoperf workload batch sizes.

### E.2. Transformer-specific experiments

In this subsection, we comment on the experiments detailed in Section 3.

### E.2.1. DATASET AND TOKENIZER

For the experiments in Section 3, we train on the C4 dataset (Raffel et al., 2020), using use the SentencePiece tokenizer from Raffel et al. (2020).

### E.2.2. ARCHITECTURE

We trained a standard decoder-only transformer from the Nanodo codebase (Liu et al., 2024), using TPU v5e. The architecture consists in $N$ identical transformer blocks. Each block consists in a multi-head causal dot-product self-attention layer followed by a MLP layer. Layer normalization is applied to both the input of the attention and the input of the MLP layer. The MLP layer uses a gelu activation function. The model employs a learned positional embedding. The model uses an independent layer to map back to the vocabulary space. So in plain code, the architecture reads

```
def Transformer(V, L, D, H, F)(x_L):
    x_LxD = Embedder(D)(x_L)
    y_LxD = PositionalEmbedder(D)([0, ...,L])
    x_LxD = x_LxD + y_LxD
    for i in range(N):
        y_LxD = LayerNorm()(x_LxD)
        y_LxD = MultiHeadAttention(H)(y_LxD)
        x_LxD = y_LxD + x_LxD
        y_LxD = LayerNorm()(x_LxD)
        y_LxD = MLP(F)(y_LxD)
        x_LxD = y_LxD + x_LxD
    x_LxD = LayerNorm()(x_LxD)
    x_LxV = Linear(V)(x_LxV)
    return x_LxV

def MLP(F)(y_LxD):
    y_LxF = Linear(F)(y_LxV)
    y_LxF = gelu(y_LxF)
    y_LxD = Linear(D)(y_LxF)
    return y_LxD
```

The key parameters are given below. The number of the backbone parameters (i.e., ignoring embedding and output layers) is 151 million.

1. the size of the vocabulary $V$, fixed by the C4 dataset (i.e., $V = 32101$)
2. the length of the sequences $L = 2024$
3. the number of blocks, $N = 12$
4. the hidden dimension $D = 64H$
5. the number of heads in the attention layer $H = 16$
6. the hidden dimension in the MLP $F = 4D$

For the memory comparisons in Figure 1, we used the same architecture but with $N = 24$, $H = 32$.

### E.2.3. OPTIMIZATION GENERIC DETAILS

Below, we present generic hyperparameters' settings. Specific setups are detailed in E.2.4.

**Number of steps.** In all experiments, the number of steps is fixed as (following Nanodo's recipe (Liu et al., 2024) which follows Chinchilla's (Hoffmann et al., 2022) scaling factor $c = 20$)

$$K = \lfloor cP/T \rfloor,$$

where $P = 12ND^2 + VD$ is the number of parameters (ignoring the final head), and $T = LB$ is the number of tokens per batch.

**Batch size.** If not specified the batch size is fixed at

$$B = 64.$$

For the memory comparison in Figure 1, we used $B = 256$.

**Schedule and peak learning rate.** We use a cosine decay schedule with warmup starting at 0, ending at 0. We use 1000 steps for the warmup. If not specified, the learning rate is set to the base learning rate of Nanodo, i.e.

$$\eta = 2/1024.$$

**Weight decay.** All implementations use weight decay, that is all implementations of ADAM and all the SIGNSGD implementations. The weight decay is fixed at

$$\omega = 8/K$$

for $K$ the number of steps presented above. This weight decay is not multiplied by the varying learning rate.

E.2.4. EXPERIMENTAL SPECIFIC DETAILS.

.

**SIGNSGD.** For SIGNEMA, we used an EMA with decaying parameter $\beta = 0.9$. The learning rates were searched around the base learning rate $\eta$ defined above in a grid $\{10^{0.25i}\eta \mid -6 \le i \le 1\}$. The best learning rates found were $3.47 \cdot 10^{-3}$ for MICROSIGNSGD, $1.10 \cdot 10^{-3}$ for SIGNSGD, $3.47 \cdot 10^{-4}$ for SIGNEMA.

**ADAM and MICROADAM.** In all experiments, we ran ADAM and its variants with EMA parameters $\beta_1 = 0.9$ and $\beta_2 = 0.95$ for respectively the first and second moments running estimators

For the scaling experiments (Figure 5, 5), as the batch size varies, the total number of steps vary according to the rule $K = \lfloor cP/T \rfloor$ with $T = LB$. The number of warmup steps is multiplied by $B/1024$ as it had been selected for batch size of 1024. As the base learning rate was chosen for a batch size $B = 1024$, the learning rate is adjusted as $\eta_\gamma = \gamma\eta$ with a factor $\gamma = B/1024$ for all variants except ADAM that uses $\gamma = \sqrt{B/1024}$. This same factor is used to adjust the weight decay as $\omega_B = \omega_\gamma$.

Although not reported here, we tried varying $\beta_2$ for MICROADAM without success.

**MICROADAMVAR algorithm** We implemented MICROADAMVAR by using two EMA estimators $\nu_{\mathrm{adam}}$ and $\nu_{\mathrm{micro}}$, and combining them in accordance with Equation 6. We then carried out a similar batch size scaling experiment to Figure 5.

# F. Detailed pseudocode

For completeness we detail the pseudocode for the MICROADAM family of algorithms in Algorithm 1.

---

**Algorithm 1** Various ADAM algorithms

---

1: **Inputs**
2:    Initial parameters $\theta_0$,
3:    Exponential Moving Average (EMA) decay parameters $\beta_1, \beta_2$
4:    Learning rate schedule $(\eta_k)_{k \geq 0}$
5:    Small $\varepsilon$ to avoid division by $0$
6:    Adam variant ADAMVARIANT
7: **Initialization**
8:    EMA of gradients $m_0 = 0$
9:    EMA of preconditioner $v_0 = 0$
10: **for** $t \in \{1, ..., T\}$ **do**
11:    Fetch samples $x_1^{(t)}, \ldots, x_B^{(t)}$
12:    Compute average gradient
13:      $\mu_t = \frac{1}{B} \sum_{i=1}^{B} g_{i,t}$ for $g_{i,t} = \nabla f(\theta_t; x_i^{(t)})$
14:    Update EMA of gradients
15:      $m_t = \beta_1 m_{t-1} + (1 - \beta_1)\mu_t$
16:    **if** ADAMVARIANT is ORIGINAL **then**
17:      Compute preconditioner as square of average gradients
18:      $\nu_t = \mu_t^2 = \left( \frac{1}{B} \sum_{i=1}^{B} g_{i,t} \right)^2$
19:    **else if** ADAMVARIANT is MICROADAM **then**
20:      Compute preconditioner as average element-wise square
21:      $\nu_t = \frac{1}{B} \sum_{i=1}^{B} g_{i,t}^2$
22:    **else if** ADAMVARIANT is MICROADAMVAR **then**
23:      Compute preconditioner as element-wise estimate variance
24:      $\nu_t = \frac{B}{1-B} \left( \frac{1}{B} \sum_{i=1}^{B} g_{i,t}^2 - \left( \frac{1}{B} \sum_{i=1}^{B} g_{i,t} \right)^2 \right)$
25:    **else if** ADAMVARIANT is MICROADAMMSQ **then**
26:      Compute preconditioner as element-wise estimate squared expectation
27:      $\nu_t = \frac{B}{1-B} \left( \left( \frac{1}{B} \sum_{i=1}^{B} g_{i,t} \right)^2 - \frac{1}{B} \sum_{i=1}^{B} g_{i,t}^2 \right)$
28:    **end if**
29:    Update EMA of preconditioner
30:      $v_t = \beta_2 v_{t-1} + (1 - \beta_2)\nu_t$
31:    Apply bias corrections on both EMAs:
32:      $\hat{m}_t = m_t/(1 - \beta_1^t)$,
33:      $\hat{v}_t = v_t/(1 - \beta_2^t)$
34:    **if** MICROADAMX = MICROADAMMSQ **then**
35:      $\hat{v}_t \leftarrow \max(0, \hat{v}_t)$
36:    **end if**
37:    Define update direction $u_t = \hat{m}_t/\sqrt{\varepsilon + \hat{v}_t}$
38:    Apply update $\theta_t = \theta_{t-1} - \eta_t u_t$
39: **end for**

---

