# OpenReview forum: "Per-example Gradients: a New Frontier for Understanding and Improving Optimizers"
_ICML.cc/2026/Conference — ICML 2026 regular_

### Official Review · Reviewer_Bpnz · 2026-03-09

**Soundness:** 3
**Presentation:** 2
**Significance:** 3
**Originality:** 3
**Overall Recommendation:** 5
**Confidence:** 2

**Summary:**

This paper challenges the view that “computing generic gradient statistics is not prohibitively expensive” by studying the technical challenges arising from computing generic gradient statistics. It is shown that staged programming languages enable generic manipulations of mini-batch gradient computations, and the implementation of per-example or per-token operations with negligible computational or memory overhead is also presented. These findings could be used to re-examine optimization algorithms, and this paper uses gradient statistics to improve the understanding of two non-linear optimization algorithms: signSGD and Adam variants. This paper shows that per-example gradient information unlocks new avenues for algorithm analysis and design.

**Compliance With Llm Reviewing Policy:**

Affirmed.

**Final Justification:**

My major concerns have been addressed. I intend to raise my score to 5.

**Key Questions For Authors:**

1. In Section 3, the authors present how to use gradient statistics to improve our understanding of two optimization algorithms, signSGD and Adam. Does the analysis here for two algorithms also provide some insights into the understanding of other algorithms?


2. On Line 430, the authors mention that, given the results in this paper, it is possible to test ideas about manipulating distributions of gradients. Can you provide more detail on this point?

**Limitations:**

The authors do not discuss the limitations of this paper, while some possible application of the findings in this paper are discussed in Section 4.

**Strengths And Weaknesses:**

Strength:
1. This paper studies the technical challenges of computing generic gradient statistics. It is shown that, in some workloads, staged programming languages enable generic manipulations of mini-batch gradient computations. The implementation of per-example or per-token operations with negligible computational overhead is presented.

2. This paper uses gradient statistics to improve the understanding of two non-linear optimization algorithms, signSGD and Adam variants, and opens a new dimension for understanding and improving algorithms.

Weakness:
1. A more comprehensive analysis of related work could be given with a detailed background in this field. Currently, only limited details are provided for the most relevant papers, including those about efficiently computing statistics per-example gradient norms.

2. The abstract could be more concise and should fit the suggested length of 4 to 6 sentences. The structure of the abstract could be clearer, it may be confusing that ‘Finally’ is followed by ‘First’ and ‘Second’.

Also see the Question part.

---

> ### Author Rebuttal · Authors · 2026-03-27
>
> We thank the reviewer for the time and the consideration they gave to this paper. Below are answers to their comments and questions.
>
> > Does the analysis here for two algorithms also provide some insights into the understanding of other algorithms?
>
>
> For signSGD, we believe a similar conclusion would hold for other normalizations, like spectral normalizations (muon). For Adam, our conclusions probably extend to rmsprop-like optimizers. Most importantly, we show clearly how accessing variance metrics help diagnose an algorithm. We believe that this is a universal fact. Variance of the gradients in the mini-batch tells a lot about e.g. whether the gradients are coherent, whether the local landscape is sharp (as a byproduct of sharpness) etc… A ratio of variance to gradient norm is for example used to assess the critical batch-size by [1].
>
> > the authors mention that, given the results in this paper, it is possible to test ideas about manipulating distributions of gradients. Can you provide more detail on this point?
>
> Consider one entry of a gradient. Rather than taking its average in the minibatch we may remove the 10% smallest and largest values and then average. Similarly Zielinski et al., (2020) studied how we can get a geometric median of the batch rather than the average and observed some gains. That paper was based on the idea that we may want more "coherent gradients". By using the leeway we found in transformers to use per-example gradient transformations, we may take a second look at these methods that changed the distribution of gradients they were looking at.
>
>
> [1] McCandlish, S., Kaplan, J., Amodei, D. and Team, O.D., 2018. An empirical model of large-batch training. arXiv preprint arXiv:1812.06162.

---

> > ### Author Rebuttal · Reviewer_Bpnz · 2026-04-02
> >
> > Thanks for your response.

---

### Official Review · Reviewer_5XuF · 2026-03-12

**Soundness:** 3
**Presentation:** 4
**Significance:** 4
**Originality:** 4
**Overall Recommendation:** 6
**Confidence:** 4

**Summary:**

The authors point out that statistics of per-exmaple gradients is under-explored due to the memory requirement of the current implementational architecture of the practical frameworks like PyTorch, Tensorflow, etc. The authors hack the JAX framework to get the per-example gradient statistics for a limited set of optimizers for negligible increase in computational overhead, and explores the meanings of the per-example gradients.

**Compliance With Llm Reviewing Policy:**

Affirmed.

**Final Justification:**

My concerns have been addressed. I maintain my score.

**Key Questions For Authors:**

1. I find this nice to read, and I find no drastic flaws that prohibit acceptance of this paper. Nice work. Well done, authors.
2. However, if we can go further generalizing this findings as I have suggested in the weaknesses/suggestion section in the above, please answer to these questions.

**Limitations:**

Yes.

**Strengths And Weaknesses:**

### Strengths

1. (Presentation) The paper is very nicely written. Reading flows well without blockings, and the authors have paid extreme care on the delivery of their idea through equations, diagrams, and organization of the paper.
2. (Significance) The paper tracks down the real practical implementation units of the automatic differentiation framework, finds and updates the existing modules to support overhead-free utilization of per-example gradient statistics.
3. (Presentation) The paper also contains all necessary codes to reproduce the resutls.
4. (Originality) As claimed by the authors, the hack of JAX to obtain a practical solution for per-example gradient-based optimization shows a novel and expandable avenue of optimization research and development. This also boosts the significance of this work.

---

### Weaknesses & Suggestion

1. (Soundness) This is a suggestion rather than weaknesses, but it will be perfect if the authors provide what other types of optimization algorithms the proposed practice can be applied *in general*.
2. (Presentation) A minor issue: Figure 8 is not quite readable.

---

> ### Author Rebuttal · Authors · 2026-03-27
>
> We thank the reviewer for the time and the consideration they gave to this paper. We really appreciate their enthusiasm for this research direction. Here is an answer to their main comment.
>
> > it will be perfect if the authors provide what other types of optimization algorithms the proposed practice can be applied in general.
>
> Generally speaking, any optimizers which carry out non-linear operations on their gradients are in-scope for the per-example methods. For example, gradient clipping can be done per-example instead of globally.

---

> > ### Author Rebuttal · Reviewer_5XuF · 2026-04-02
> >
> > I will keep my score. Thank you for the response.

---

### Official Review · Reviewer_Vqns · 2026-03-13

**Soundness:** 3
**Presentation:** 4
**Significance:** 4
**Originality:** 3
**Overall Recommendation:** 5
**Confidence:** 3

**Summary:**

In this paper, the authors observe that most optimization algorithms typically only use mini-batch averaged gradient information to update parameters. This is because keeping track of and using arbitrary functions of per-sample/token gradients is considered computationally expensive and infeasible in modern autodifferentiation frameworks.

They then show, building on the work by Dangel et al. that it is in fact possible to access per-example gradient statistics. Specifically, they perform ‘surgery’ on the computational graph where mini-batch gradients are computed, manipulating this graph such that any factorable operation can be performed on the mini-batch gradients, before averaging. They also show that functional programming languages like JAX can be used to efficiently implement this computational graph manipulation.

Now having access to per-example statistics, the authors revisit two optimization methods, SignSGD and ADAM, and ask the question what the more detailed knowledge about the gradients can actually be used for.
For SignSGD, they study the effect of the placement of the sign operation, and find that placing it as late as possible, after maximal averaging, works best. They also provide a simple argument based on the signal-to-noise ratio explaining this behavior.
For Adam, the authors investigate the difference between averaging the square of gradients (MicroAdam), or squaring the average of all gradients. As opposed to the modern view that Adam primarily leverages variance information (square of mean), it is found that a pre-conditioner dominated by the mean squared leads to more efficient training.

Altogether, this shows that training algorithms can benefit from higher order/nonlinear gradient statistics, and are also efficiently computable for modern architectures.

**Compliance With Llm Reviewing Policy:**

Affirmed.

**Key Questions For Authors:**

For what classes of architectures exactly does the graph surgery method fail? Do these failure points occur in modern transformers?

Does the claim about the Adam preconditioner depend on hyperparameter choice, or is it robust?

Although these gradients are also available in PyTorch, this is much less natural or core to the library than it is in Jax. Given the PyTorch is the de facto standard for much of deep learning, maybe the authors could provide some additional discussion on PyTorch specifically and the current state of this library for obtaining these gradients in popular workflows (e.g. training large language models). More specifically, I think a camera ready version of this paper would benefit from being very clear to an unfamiliar reader up front about exactly when per sample gradients might be feasible or not.

**Limitations:**

Mostly yes, but the discussion lacks a limitation sentence/section, and the improvement over current methods (Dangel et al.) can use an additional sentence, as the work seems to build on this quite strongly.

**Strengths And Weaknesses:**

Strengths:
* The observation that the computational graph structure of mini-batch gradients often preserves per-example information until a final reduction aligns with the idea of the computational graph surgery method and makes perfect sense.
* The feasibility result that the surgery method does not necessarily require more memory and allows for low-overhead computation of gradient statistics is important in the light of experimenting with these ideas and insights in very large modern settings.
* The authors perform experiments in large-model regimes, relevant to modern deep learning architectures
* The case study considers widely used optimizers like Adam, increasing impact and the chance of transfer of these insights to similar optimizer built on this concept.
* Making training algorithms more efficient is crucial in the light of energy consumption, hence developing techniques that enable research into these algorithms, such as this paper does, is highly significant.

Weaknesses
* The notation that the authors refer to as ‘Einstein summation convention’ is not how it was meant and is commonly used in physics, where repeated indices are contracted and summed over. This does not change any of the results, but might be confusing to some readers.
* The authors build on the work by Dangel et al., which already claims to have already solved the problem of efficiently gaining additional information about first and second order gradients, and introduced a PyTorch framework, BackPACK, to do this. Although the application to signSGD and Adam clearly provides novel insights, it is not entirely clear why the method by Dangel et al. wasn’t sufficient already, and why the additional computational graph surgery method is necessary. The authors write that ‘general development remains difficult’, but provide no clear example/citation of why it remains difficult.
* Some conclusions, such as that mean-dominated preconditioners are better than variance-dominated preconditioners, might be explained partially by choices of training parameters, e.g. learning-rate scaling, clipping, warmup schedules, etc. It is thus not entirely clear if the claim can be made so general, or if it is more sensitive to hyperparameters.
* The language used in section 2 on the computational graph surgery might not be familiar to all readers, and a very brief introductory sentence explaining the notation $A ← B, C$ might be helpful, allowing more people to easily read the paper and use its results.
In the discussion, section 4, the authors only highlight the interesting parts of the research, and do not at all mention limitations or other weaknesses, such as for how many architectures the low computational overhead surgery methods actually applies



Soundness - Because the experiments are well-designed to support the computational efficiency claims, making the results sound, but the generality of the result in terms of when it works (for which architectures exactly can the surgery method be employed efficiently?) and its applicability (only two cases, SignSGD and Adam are studied, one more would be ideal) could be stronger.

Presentation - Very well-structured paper and readable, with plenty of detailed background in the appendices to potentially reproduce results and use in further research. However, as pointed out the Einstein convention is used in a strange manner, and the notation in section 2 could use a little bit of clarification. Additionally, the exact increment over Dangel et al. when it come to efficiency is unclear.

Significance - Making training algorithms more efficient is crucial in the light of energy consumption, hence developing techniques that enable research into these algorithms, such as this paper does, is highly significant.

Originality - Either fair or good, the work by Dangel et al. has nearly the exact same motivation in its abstract, and it is unclear to me exactly how applicable the computation graph surgery method is in slightly different settings (e.g. attention layers). Could be good, because the authors do find a surprising result on Adam, opposing modern views, implying original research.


I include the below informal notes to assist the authors in in preparing a camera ready version if the paper is accepted:

Minor typos:
* “There have been attempts to efficiently compute statistics of per-example gradient norms”
* “Our results suggest that computing generic gradient statistics is not prohibitively expensive, and in some cases comes at virtually no overhead.”
* “​​It illustrates that experimentation with per-example methods on transformers is not prohibitively expensive”

Comments while reading
* Maybe elaborate a bit on what is new w.r.t. Dangel, since they build on this work, which already seems to implement efficient per-example gradient computations
* “general development remains difficult.” why does it remain difficult in these approaches?
* “On specific architecture families” How specific?
* “In contrast to conventional wisdom” citation for this claim
* Notation below fact 2.1 is new to me, e.g. (D,DF→F) and not immediately clear to me what is meant (I understand it, but might be helpful to add a brief sentence on notation)
* Not super important, but authors do not state which dataset is used for the transformer experiment in Fig. 1?
* “May be able to automatically identify and exploit this structure in some architectures” which architectures?
* Why log-scale in figure 2? Seems a bit unnecessary given the axis range (~0 to 30), but I can understand that it looks better, not a big point
* Fig. 2 I don’t fully understand/agree with the conclusions drawn from this figure. The authors say that “this modest increase in memory may be paid by a nonnegligible increase in computational cost” for the non-transformer models, but this seems like a speculation, if this is not the case than these models are also quite efficient with computing gradient statistics already. Maybe I misunderstand the point here.
* General misuse of Einstein summation convention (from a physics perspective at least)

---

> ### Author Rebuttal · Authors · 2026-03-27
>
> We thank the reviewer for the time and dedication they put in reviewing this paper. We are very grateful to the typos they found as well as their many comments that will improve the manuscript for its final version.
>
> > a very brief introductory sentence explaining the notation A ← B , C might be helpful
>
> Thank you for your note about Einstein summation convention. In the main text we actually use the explicit version of the `np.einsum` notation; in the appendix we generally have explicit sums when necessary. We will update the text to be more clear on this point.
>
> > it is not entirely clear why the method by Dangel et al. wasn’t sufficient already
>
> The main reason we introduced the computational graph surgery is for flexibility and usability. BackPACK requires using a dedicated library of layers in pytorch. Practitioners who want to use BackPACK must build their model out of the custom layers provided only. If you have an idea for a different layer type, you would need to write a new BackPACK layer with the custom gradient implementation.
>
> Our computational graph surgery in JAX works at the level of the jaxpr — effectively, one level below the numpy layer. This means that we only need to be able to handle a small number of primitives; once those are handled, any layers derived from the core operations will automatically be handled, and the user is surfaced a function like “value_grad_and_grad_var”. We believe that this is the right level of abstraction to build new oracles on functions (like accessing gradient variance), as it allows us to think in computational graph terms that isolate  potential complications. For example, this approach allows for easier extension to shared weights as we may retrieve shared weights by parsing the graph (see Appendix B.4 for a discussion). We can also easily add additional non-linear functions by modifying only relevant base primitives. For example in our paper we included experiments with the sign function (which is not included in BackPACK). The underlying code was a simple extension of the computational graph surgery framework that minimized the amount of additional engineering effort needed to define the new operation.
>
> The dream would be for the compiler to natively handle all these optimizations in HLO (see appendix A for an introduction of the journey of python code in jax); until that day, we believe the computational graph surgery approach allows for practitioners to more freely use the methods without having to rewrite their machine learning systems.
>
> > Does the claim about the Adam preconditioner depend on hyperparameter choice, or is it robust?
>
> We completely agree that claiming the superiority of MicroAdamMSQ over regular Adam requires a much more robust set of experiments; that is beyond the goal of this manuscript. With regards to our claims: we swept over the learning rates, as well as beta2 and the clipping thresholds. The stability issues in our naive per-example optimizers unfortunately persisted across a wide range of parameter settings. Our experiments on stabilizing AdamMSQ suggest that progress is possible; in future work, we hope to use the per-example measurements to better understand the instabilities and provide robust stabilization that doesn’t slow down optimization.
>
> > the authors [...] do not at all mention limitations
>
> Thank you for the feedback here. The main limitations are the inefficiency of vmap in certain architectures, and the fact that the "mathematical tricks" developed originally by Dangel et al in BackPACK may not work in some situations, see answer below. We attempted to present these issues in our main text but clearly failed. We will edit the main text to emphasize these points.
>
> > For what classes of architectures exactly does the graph surgery method fail?
>
> The main question is: when do tricks like inserting the square into the inputs be implemented or not? The main problematic cases are weight tying and preprocessed weights; see Appendix B.4 for a more detailed discussion. Generally, any case that breaks the low rank structure or the locality of the computations makes the methods worse. The computational graph surgery may fall back on simpler methods in these cases. The only failure that might occur in modern transformers is weight-tying of the embedding layer, but this only modestly degrades performance. We note that the vmap methods work “well enough” for many well studied architectures to be used for measurements.
>
> > Although these gradients are also available in PyTorch, this is much less natural or core to the library than it is in Jax...
>
> Good point. Pytorch also has a vmap functionality. We will add a timing of micro-adam vs adam in pytorch to give an idea to the user. For a "computational graph surgery" approach we will point the reader to BackPACK. We really believe that JAX offers the best possible abstraction for future research and for our own current development.
>
> Thank you again for the thorough review.

---

> > ### Author Rebuttal · Reviewer_Vqns · 2026-04-03
> >
> > I thank the authors for their response. I maintain my original positive assessment of the paper. I leave the suggested improvements to the discretion of the authors in how they should feature in a camera ready version.

---

### Official Review · Reviewer_5GfE · 2026-03-15

**Soundness:** 3
**Presentation:** 3
**Significance:** 3
**Originality:** 2
**Overall Recommendation:** 5
**Confidence:** 3

**Summary:**

This paper challenges the traditional view behind standard deep learning training algorithms that computing per-example gradient statistics is prohibitively expensive. They show that for sequence-level architectures (like Transformers), the memory bottleneck is actually in activation checkpoints rather than individual gradients; using automatic vectorization tools, like vmap in JAX, enables computing per-example gradients without increasing peak memory usage. In addition, they also explore how access to per-example gradients can enable new avenues for algorithm analysis and design. First, they study variants of SignSGD, depending on the position of EMA, sign, and avg operations, finding that SignEMA (sign after EMA) trains best. They support this with theoretical analysis to show that applying sign after batch averaging is superior to applying it to per-example gradients. Second, they study a variant of Adam, namely MicroAdam, which uses the average element-wise squared gradients over a batch for the preconditioner (rather than the squared average gradients). Previous work studied an approximate version of MicroAdam and derived a batch size scaling of the learning rate as linear (vs square root for Adam), which this paper verifies holds for the exact version. However, they find that using Adam performs better in settings where the mean-squared term dominates the effective variance, which they verify through measurements enabled by their framework. However, this leaves open the question of the effectiveness of the batch-scaling scaling of learning rate for Adam (which seems to assume the preconditioner is variance dominated).

**Compliance With Llm Reviewing Policy:**

Affirmed.

**Final Justification:**

I maintain my positive assessment of the paper.

**Key Questions For Authors:**

It would be helpful to include some discussion on [1] in the discussion section.

Minor suggestion to improve readability: line 317 seems to be missing the word 'consider'.

References:

[1] Seesaw: Accelerating Training by Balancing Batch Size and Learning Rate Scheduling, ICLR 2026.

**Limitations:**

Yes.

**Strengths And Weaknesses:**

**Strengths**:

The paper studies an important and interesting angle to use per-example gradients to get further insight into optimizer algorithm analysis and design.

The methodological contribution to develop computational graph surgery to inject non-linear operations (like squaring or sign) into the gradient computation before batch averaging, as well as the analysis and findings, such as comparing Adam preconditioners and showing that a preconditioner dominated by the mean squared gradient leads to more stable and faster training than one dominated by variance, are interesting and valuable.

The paper is very well-written overall.

**Weaknesses**:

The paper does not have any major weaknesses, some minor ones are as follows.

It focuses mainly on SignSGD and Adam variants; it is not explore how these per-example techniques can provide similar benefits to other classes of optimizers.

While the analysis on potential benefits of using per-example gradients is interesting, the paper does not present a directly useful alternative optimizer/scheduler based on the analysis. That would improve the paper, but I don't believe it's absolutely necessary for the paper.

---

> ### Author Rebuttal · Authors · 2026-03-27
>
> We thank the reviewer for the time and the consideration they gave to this paper. Below are answers to their comments and questions.
>
> > the paper does not present a directly useful alternative optimizer/scheduler based on the analysis.
>
> This is a fair comment. Micro-Adam was our initial attempt at a better algorithm, though our results suggest that AdamMSQ is a more promising candidate. We believe that the measurement tools provided by our per-example techniques can actually be even more useful than a single new optimizer/scheduler. We demonstrated how these tools can be used to understand and improve these variants of Adam to make them viable to practitioners.
>
> > It would be helpful to include some discussion on [1] Seesaw: Accelerating Training by Balancing Batch Size and Learning Rate Scheduling, ICLR 2026
>
> Thanks for bringing this to our attention. This paper uses the usual assumption that "variance dominates the mean squared in the batch" to develop a new learning rate and batch size scaling rule. As for the original "square root learning rate scaling rule" we do not dispute its success but clearly the underlying assumption may not be true in practice and there may be better empirically founded assumptions to make here. We'll add such a discussion in the conclusion.
>
> We thank the reviewer for pointing out the typo, we'll correct it.

---

> > ### Author Rebuttal · Reviewer_5GfE · 2026-04-01
> >
> > Thank you for the response. I will maintain my score.

---

### Decision · Program_Chairs · 2026-04-30

**Decision:**

Accept (regular)

**Comment:**

This paper shows that per-example gradient statistics can be computed efficiently and leverages this to provide new insights into optimizer design, particularly for signSGD and Adam variants. Reviewers agree that the paper is technically sound, well-written, and addresses an important problem. The main concerns are limited scope (focused on a small set of optimizers) and some lack of clarity on novelty over prior work. These issues are relatively minor and were largely addressed in the rebuttal. Overall, given the positive consensus from reviewers, I recommend acceptance.